# Single-molecule tracking in live cells reveals distinct target-search strategies of transcription factors in the nucleus

Ignacio Izeddin[1,2†], Vincent Récamier[1†‡], Lana Bosanac[1,3], Ibrahim I Cissé[1,2§], Lydia Boudarene[1,2,4], Claire Dugast-Darzacq[1,4], Florence Proux[1], Olivier Bénichou[5], Raphaël Voituriez[5], Olivier Bensaude[6], Maxime Dahan[2,3,7*], Xavier Darzacq[1,3*¶]

[1]Functional Imaging of Transcription, Institut de Biologie de l'Ecole Normale Supérieure (IBENS), Inserm U1024, and CNRS UMR 8197, Paris, France; [2]Laboratoire Kastler Brossel, CNRS UMR 8552, Departement de Physique et Institut de Biologie de l'Ecole Normale Supérieure (IBENS), Paris, France; [3]Transcription Imaging Consortium, Janelia Farm Research Campus, Howard Hughes Medical Institute, Ashburn, United States; [4]Université Paris Diderot, Paris, France; [5]Laboratoire de Physique Théorique de la Matière Condensée, CNRS UMR 7600, Université Pierre et Marie Curie, Paris, France; [6]Cell Biology of Transcription, Institut de Biologie de l'École Normale Supérieure (IBENS) CNRS UMR 8197, Paris, France; [7]Laboratoire Physico-Chimie, Institut Curie, CNRS UMR 168, Paris, France

*For correspondence: maxime.dahan@curie.fr (MD); darzacq@ens.fr (XD)

†These authors contributed equally to this work

Present address: ‡Laboratory Imaging, s.r.o., Prague, Czech Republic; §Department of Physics, Massachusetts Institute of Technology, Cambridge, United States; ¶Division of Genetics, Genomics & Development, Department of Molecular and Cell Biology, University of California, Berkeley, Berkeley, United States

Competing interests: The authors declare that no competing interests exist.

**Abstract** Gene regulation relies on transcription factors (TFs) exploring the nucleus searching their targets. So far, most studies have focused on how fast TFs diffuse, underestimating the role of nuclear architecture. We implemented a single-molecule tracking assay to determine TFs dynamics. We found that c-Myc is a global explorer of the nucleus. In contrast, the positive transcription elongation factor P-TEFb is a local explorer that oversamples its environment. Consequently, each c-Myc molecule is equally available for all nuclear sites while P-TEFb reaches its targets in a position-dependent manner. Our observations are consistent with a model in which the exploration geometry of TFs is restrained by their interactions with nuclear structures and not by exclusion. The geometry-controlled kinetics of TFs target-search illustrates the influence of nuclear architecture on gene regulation, and has strong implications on how proteins react in the nucleus and how their function can be regulated in space and time.

## Introduction

The nucleus is a complex environment where biochemical reactions are spatially organized in an interaction network devoted to transcription, replication, or repair of the genome (*Misteli, 2001*). Molecular interactions relevant to gene regulation involve transcription factors (TFs) that bind to specific DNA regulatory sequences or other components of the transcriptional machinery. In order to find their targets, TFs diffuse within the seemingly non-compartmentalized yet highly organized nuclear volume. Since the kinetics of a reaction can be largely determined by the mobility characteristics of the reactants (*Rice, 1985*; *Shlesinger and Zaslavsky, 1993*), the target-search strategy of TFs is a key element to understand the dynamics of transcriptional activity and regulation.

Over the past decade, the nuclear dynamics of TFs has become an important topic of research and has been investigated with a variety of imaging and biochemical approaches. Overall, these studies have emphasized the high mobility of nuclear factors, which results from a combination of diffusive motion and transient specific and non-specific interactions with chromatin (*Darzacq et al., 2009*;

**eLife digest** Transcription factors are proteins that control the expression of genes in the nucleus, and they do this by binding to other proteins or DNA. First, however, these regulatory proteins need to overcome the challenge of finding their targets in the nucleus, which is crowded with other proteins and DNA.

Much research to date has focused on measuring how fast proteins can diffuse and spread out throughout the nucleus. However these measurements only make sense if these proteins have access to the same space within the nucleus.

Now, Izeddin, Récamier et al. have developed a new technique to track single protein molecules in the nucleus of mammalian cells. A transcription factor called c-Myc and another protein called P-TEFb were tracked and while they diffused at similar rates, they 'explored' the space inside the nucleus in very different ways.

Izeddin, Récamier et al. found that c-Myc explores the nucleus in a so-called 'non-compact' manner: this means that it can move almost everywhere inside the nucleus, and has an equal chance of reaching any target regardless of its position in this space. P-TEFb, on the other hand, searches the nucleus in a 'compact' way. This means that it is constrained to follow a specific path through the nucleus and is therefore guided to its potential targets.

Izeddin, Récamier et al. explain that the different 'search strategies' used by these two proteins influence how long it takes them to find their targets and how far they can travel in a given time. These findings, together with information about where and when different proteins interact in the nucleus, will be essential to understand how the organization of the genome within the nucleus can control the expression of genes. The next challenge will now be to uncover what determines a protein's search strategy in the nucleus, as well as the potential ways that this strategy might be regulated.

*Mueller et al., 2010*; *Normanno et al., 2012*). These transient interactions are essential to ensure a fine regulation of binding site occupancy—by competition or by altering the TF concentration—but must also be persistent enough to enable the assembly of multicomponent complexes (*Dundr, 2002*; *Darzacq and Singer, 2008*; *Gorski et al., 2008*; *Cisse et al., 2013*).

In parallel to the experimental evidence of the fast diffusive motion of nuclear factors, our understanding of the intranuclear space has evolved from a homogeneous environment to an organelle where spatial arrangement among genes and regulatory sequences play an important role in transcriptional control (*Heard and Bickmore, 2007*). The nucleus of eukaryotes displays a hierarchy of organized structures (*Gibcus and Dekker, 2013*) and is often referred to as a crowded environment.

How crowding influences transport properties of macromolecules and organelles in the cell is a fundamental question in quantitative molecular biology. While a restriction of the available space for diffusion can slow down transport processes, it can also channel molecules towards their targets increasing their chance to meet interacting partners. A widespread observation in quantitative cell biology is that the diffusion of molecules is anomalous, often attributed to crowding in the nucleo-plasm, cytoplasm, or in the membranes of the cell (*Höfling and Franosch, 2013*). An open debate remains on how to determine whether diffusion is anomalous or normal (*Malchus and Weiss, 2009*; *Saxton, 2012*), and the mechanisms behind anomalous diffusion (*Saxton, 2007*). The answer to these questions bears important consequences for the understanding of the biochemical reactions of the cell.

The problem of diffusing molecules in non-homogenous media has been investigated in different fields. Following the seminal work of *de Gennes (1982a)*, (*1982b)* in polymer physics, the study of diffusivity of particles and their reactivity has been generalized to random or disordered media (*Kopelman, 1986*; *Lindenberg et al., 1991*). These works have set a framework to interpret the mobility of macromolecular complexes in the cell, and recently in terms of kinetics of biochemical reactions (*Condamin et al., 2007*). Experimental evidence has also been found, showing the influence of the glass-like properties of the bacterial cytoplasm in the molecular dynamics of intracellular processes (*Parry et al., 2014*). These studies demonstrate that the geometry of the medium in which

diffusion takes place has important repercussions for the search kinetics of molecules. The notion of *compact* and *non-compact* exploration was introduced by *de Gennes (1982a)* in the context of dense polymers and describes two fundamental types of diffusive behavior. While a *non-compact* explorer leaves a significant number of available sites unvisited, a *compact* explorer performs a redundant exploration of the space. In chemistry, the influence of compactness is well established to describe dimensional effects on reaction rates (*Kopelman, 1986*).

In this study, we aim to elucidate the existence of different types of mobility of TFs in the eukaryotic nucleus, as well as the principles governing nuclear exploration of factors relevant to transcriptional control. To this end, we used single-molecule (SM) imaging to address the relationship between the nuclear geometry and the search dynamics of two nuclear factors having distinct functional roles: the proto-oncogene c-Myc and the positive transcription elongation factor (P-TEFb). c-Myc is a basic helix-loop-helix DNA-binding transcription factor that binds to E-Boxes; 18,000 E-boxes are found in the genome, and c-Myc affects the transcription of numerous genes (*Gallant and Steiger, 2009*). Recently, c-Myc has been demonstrated to be a general transcriptional activator upregulating transcription of nearly all genes (*Lin et al., 2012*; *Nie et al., 2012*). P-TEFb is an essential actor in the transcription regulation driven by RNA Polymerase II. P-TEFb is a cyclin-dependent kinase, comprising a CDK9 and a Cyclin T subunit. It phosphorylates the elongation control factors SPT5 and NELF to allow productive elongation of class II gene transcription (*Wada et al., 1998*). The carboxy-terminal domain (CTD) of the catalytic subunit RPB1 of polymerase II is also a major target of P-TEFb (*Zhou et al., 2012*). c-Myc and P-TEFb are therefore two good examples of transcriptional regulators binding to numerous sites in the nucleus; the latter binds to the transcription machinery itself and the former directly to DNA.

Single particle tracking (SPT) constitutes a powerful method to probe the mobility of molecules in living cells (*Lord et al., 2010*). In the nucleus, SPT has been first employed to investigate the dynamics of mRNAs (*Fusco et al., 2003*; *Shav-Tal et al., 2004*) or for rheological measurements of the nucleoplasm using inert probes (*Bancaud et al., 2009*). Recently, the tracking of single nuclear factors has been facilitated by the advent of efficient in situ tagging methods such as Halo tags (*Mazza et al., 2012*). An alternative approach takes advantage of photoconvertible tags (*Lippincott-Schwartz and Patterson, 2009*) and photoactivated localization microscopy (PALM) (*Betzig et al., 2006*; *Hess et al., 2006*). Single particle tracking PALM (sptPALM) was first used to achieve high-density diffusion maps of membrane proteins (*Manley et al., 2008*). However, sptPALM experiments have typically been limited to proteins with slow mobility (*Manley et al., 2008*) or those that undergo restricted motions (*Frost et al., 2010*; *English et al., 2011*). Recently, by inclusion of light-sheet illumination, it has been used to determine the binding characteristics of TFs to DNA (*Gebhardt et al., 2013*).

In this study, we developed a new sptPALM procedure adapted for the recording of individual proteins rapidly diffusing in the nucleus of mammalian cells. We used the photoconvertible fluorophore Dendra2 (*Gurskaya et al., 2006*) and took advantage of tilted illumination (*Tokunaga et al., 2008*). A careful control of the photoconversion rate minimized the background signal due to out-of-focus activated molecules, and we could thus follow the motion of individual proteins freely diffusing within the nuclear volume. With this sptPALM technique, we recorded large data sets (on the order of $10^4$ single translocations in a single imaging session), which were essential for a proper statistical analysis of the search dynamics.

We applied our technique to several nuclear proteins and found that diffusing factors do not sense a unique nucleoplasmic architecture: c-Myc and P-TEFb adopt different nuclear space-exploration strategies, which drastically change the way they reach their specific targets. The differences observed between the two factors were not due to their diffusive kinetic parameters but to the geometry of their exploration path. c-Myc and our control protein, 'free' Dendra2, showed free diffusion in a three-dimensional nuclear space. In contrast, P-TEFb explored the nuclear volume by sampling a space of reduced dimensionality, displaying characteristics of exploration constrained in fractal structures. The role of the space-sampling mode in the search strategy has long been discussed from a theoretical point of view (*de Gennes, 1982a*; *Kopelman, 1986*; *Lindenberg et al., 1991*). Our experimental results support the notion that it could indeed be a key parameter for diffusion-limited chemical reactions in the closed environment of the nucleus (*Bénichou et al., 2010*). We discuss the implications of our observations in terms of gene expression control, and its relation to the spatial organization of genes within the nucleus.

## Results

### Intracellular single-molecule tracking with photoconvertible fluorescent proteins

We developed a simple and versatile approach based on photoconvertible protein tags that extends the use of sptPALM to any protein expressed in mammalian cells. Proteins of interest were fused to the photoconvertible protein Dendra2 (*Gurskaya et al., 2006*; *Figure 1A*). A standard wide-field configuration of the microscope allowed fast and sensitive acquisition with an EMCCD camera ('Materials and methods—Single-molecule imaging and Detection and tracking of single molecules', *Figure 1—figure supplement 1*). We used low activation intensity and tilted illumination (*Figure 1B*) in order to reach the regime of SM detection, characterized by single-step activation and photobleaching (*Figure 1C*). Due to activation of out-of-focus fluorophores, a decreasing density of detected particles was correlated with an increasing average signal-to-noise ratio (SNR) (*Figure 1D*). We found that activation intensity around 0.01 kW/cm$^2$ offered the best trade-off between the number of detected particles (~1) and SNR.

Compared to membrane proteins or other proteins with constrained mobility, diffusion dynamics of intracellular molecules is much higher and can exceed 10 μm$^2$/s. Images recorded for such fast moving objects depart from the well-defined point spread function (PSF) of the microscope and exhibit a motion blur that cannot be characterized with standard Gaussian localization algorithms (*Thompson et al., 2002*). Therefore, we developed new localization and tracking algorithms ('Materials and methods—Detection and tracking of single molecules' and *Figure 1—figure supplements 1 and 2*) and validated them with simulations ('Materials and methods—Numerical simulations' and *Figure 1—figure supplement 3*). We could thus obtain single trajectories formed by individual translocations recorded every 10 ms. 50% of the traces were reconstructed with more than four time points, and some of them were as long as 60 consecutive translocations. The step size of single translocations ranged between tens of nanometers (limited by our localization accuracy of ~70 nm) and ~2 μm (*Figure 1E* and *Videos 1–5*). Hence, it became possible to track molecules with diffusion coefficients exceeding 10 μm$^2$/s.

### System validation using 'free' Dendra2 and histone H2B fused to Dendra2

We first investigated two limit cases relevant to protein dynamics in the nucleoplasm: Dendra2 and DNA-associated histone H2B. Dendra2 is the fluorescent label that we fused to all other proteins used in our analysis. Green fluorescent protein (GFP) has no detectable interacting partners in mammalian cells (*Trinkle-Mulcahy et al., 2008*), and we therefore considered 'free' Dendra2 as a model for freely diffusing particles due to its structural similarity with GFP. In contrast, Dendra2 fused to histone H2B (Dendra2-H2B) was expected to insert into chromatin and thus to display restricted motion.

Indeed, from a visual inspection, 'free' Dendra2 and Dendra2-H2B trajectories (*Figure 2A,B*, respectively) exhibited obvious differences. Notably, translocation histograms for 'free' Dendra2 and for Dendra2-H2B were not consistent with a single diffusing species (*Figure 2—figure supplement 1*, 'Materials and methods–Cumulative histogram analysis and mean square displacement'), thus suggesting that displacements of these molecules were more complex than anticipated. Three distinct populations were needed to fit the translocation histograms at all time intervals (*Figure 2—figure supplement 1*).

To complement our analysis of the translocation histograms, we plotted the mean square displacement (MSD) of the molecules as a function of time ('Materials and methods—Cumulative histogram analysis and mean square displacement'). For Dendra-H2B, the MSD reached a plateau after ~20 ms at ~ 0.5 μm$^2$ (*Figure 2C*), consistent with a confined motion of individual histone molecules inserted into chromatin. The MSD of 'free' Dendra2 increased regularly with time. However, it slightly deviated from the linear behavior expected for molecules undergoing normal diffusion. This was attributed to a 'population exclusion effect' due to the different defocusing rates of the various diffusive subpopulations of Dendra2.

Because of their three-dimensional motion in the nucleus, slow moving particles remained within the focal depth of observation (~0.5–1 μm) for a longer time than fast moving ones. As a result, fast diffusing molecules contributed comparatively less than the slow ones to the MSD at longer time lags. Note that this effect is inevitable for any single-molecule experiment involving more than one diffusive

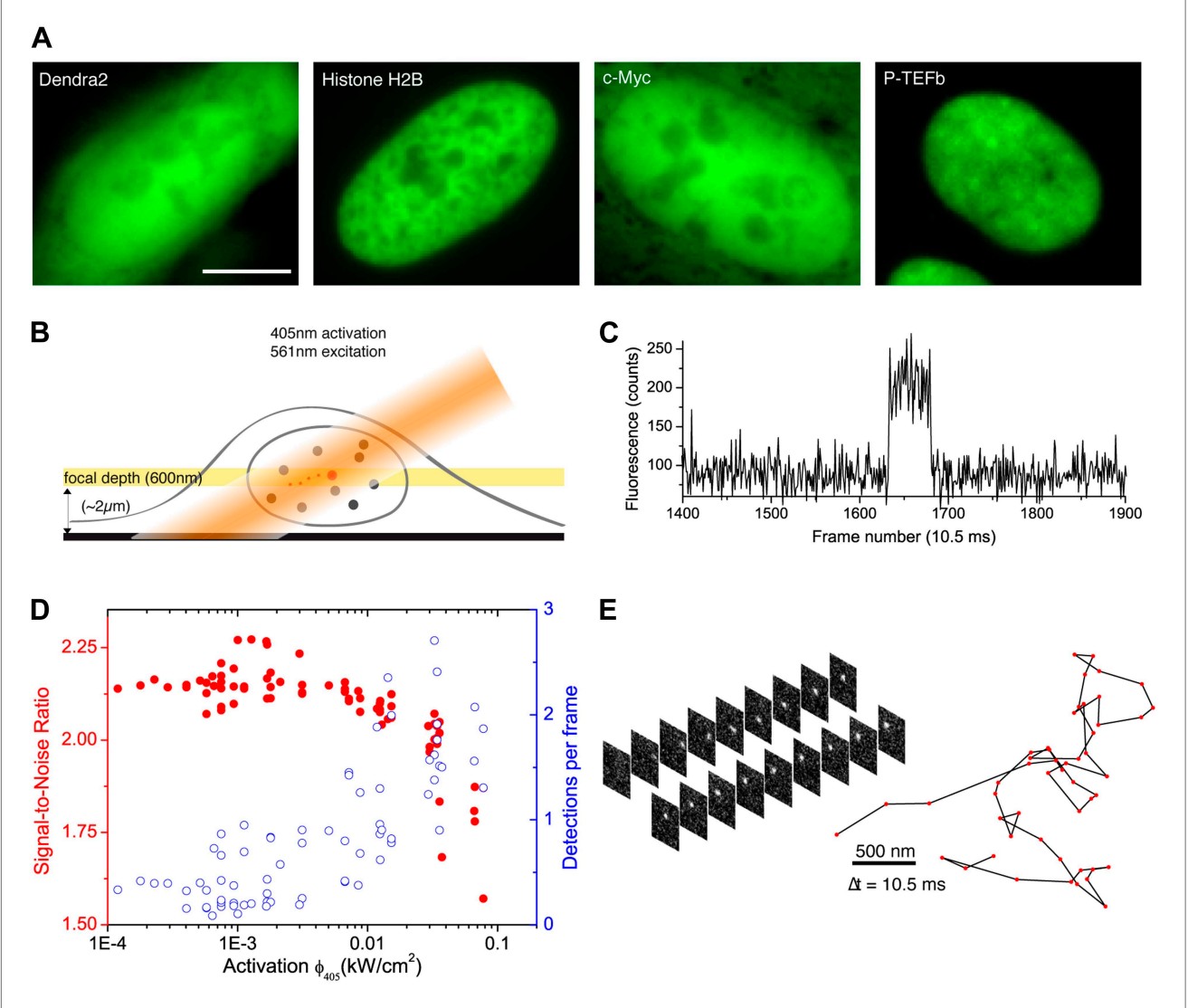

**Figure 1**. From bulk to single molecule fluorescence imaging. (**A**) Images of the 525 nm bulk emission of the pre-converted form of Dendra2 in the cellular nucleus for the 'free' fluorophore Dendra2 and Dendra2 fused to H2B, c-Myc, and P-TEFb. (**B**) Schematics of the intracellular sptPALM; wide-field illumination is necessary in order to reach the nucleus of mammalian cells. A signature of single molecule detection is the on/off single-step fluorescence shown in panel (**C**). To achieve single molecule detection, 405 nm laser photoactivation needs to be reduced to a level where no background noise is produced by out-of-focus fluorophores. Graphic in panel (**D**) shows the number of detected single molecules (blue data, right axis) and the mean SNR of the single molecule signal (red data, left axis) as a function of 405 nm photoactivation photon flux per pulse (10 ms pulses every 1 s). The signal-to-noise ratio (SNR) of the molecules within the image depth of focus indeed increases as the total number of detected particles decreases. In panel (**E**), the trace of a single Dendra2 molecule freely diffusing in the nucleus of a living cell is depicted, imaged at a rate of 95 Hz (10 ms acquisition time and 0.5 ms interval between frames).

The following figure supplements are available for figure 1:

**Figure supplement 1**. Motion blur and detection algorithm.

**Figure supplement 2**. Tracking algorithm.

**Figure supplement 3**. Localization accuracy and detection efficiency as a function of diffusion coefficient.

population and in which the three-dimensional movement of particles is recorded in two dimensions ('Materials and methods—Numerical simulations' and *Figure 2—figure supplement 2*). We therefore adjusted the rates of the different diffusive populations for each molecule, and have used the

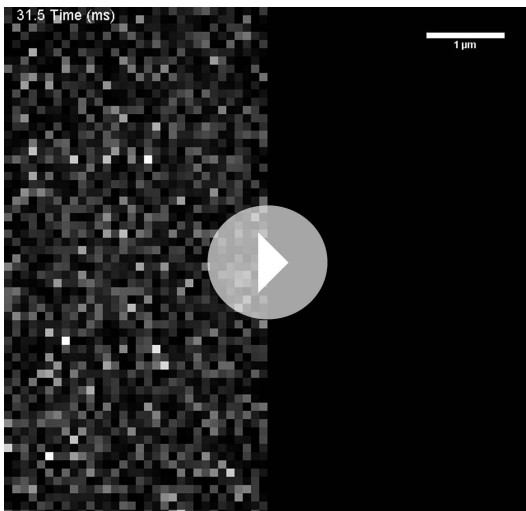

**Video 1**. Raw video of a single Dendra2 molecule diffusing in the nucleoplasm of a U2OS cell. Running parallel to the raw image, reconstruction of the trace by the localization and tracking algorithms. Exposure time was 10 ms, with 0.5 ms dead time between frames. Running time and scale bars are stamped on the video.

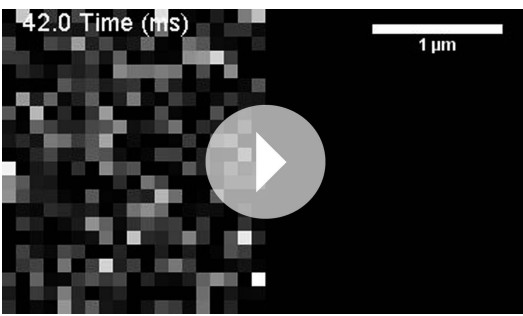

**Video 2**. Raw video of a single H2B molecule in the nucleoplasm of a U2OS cell. Running parallel to the raw image, reconstruction of the trace by the localization and tracking algorithms. Exposure time was 10 ms, with 0.5 ms dead time between frames. Running time and scale bars are stamped on the video.

corrected values through the text and for our analysis. The deviation from linearity of the MSD curve produced by such an exclusion effect clearly illustrates the need to complement the analysis of molecular mobility with other observables, ideally independent of population heterogeneity.

Finally, to carefully establish the range of application of our experimental and analytical methods, we performed numerical simulations ('Materials and methods—Numerical simulations'). On the one hand, the particle localization precision sets the lower bound to a reliable estimation of the diffusion parameters, that is ~ 0.04 $\mu m^2/s$ for a pointing accuracy of ~70 nm. On the other hand, fast moving particles can be tracked with a mobility up to ~20 $\mu m^2/s$, beyond the experimental values determined for 'free' Dendra2. Altogether, our experimental and numerical results provide a benchmark for studying nuclear factors with a mobility ranging between that of chromatin-bound H2B molecules and of 'free' proteins such as Dendra2.

## c-Myc and P-TEFb differ in the nature of their diffusion

We next probed the mobility of transcription factors. Dendra2 was fused to the proto-oncogene c-Myc and to the Cyclin T1 subunit of P-TEFb. It has recently been shown that, rather than activating new sets of genes in the cell, the role of c-Myc is that of an amplifier of transcription of already active genes (*Lin et al., 2012*; *Nie et al., 2012*). We thus tested the functionality of c-Myc-Dendra2 by performing RT-qPCR on a set of active genes in our U2OS cell line. When comparing the wild-type cells and those expressing c-Myc-Dendra2, we measured an increase of RNA expression levels in 10 out of 12 tested genes ('Materials and methods—mRNA expression and c-Myc expression amplification analysis').

Translocation histograms for c-Myc were well fit with three diffusive populations (*Figure 3—figure supplement 1*). The most abundant corresponded to rapidly diffusing particles (13.5 $\mu m^2/s$, 70% of the molecules) (*Figure 3A*, black trajectories). In addition, a significant fraction of c-Myc was immobile (9.5%) (*Figure 3A*, green trajectory) or displayed slow diffusion ($D_2 = 0.5$ $\mu m^2/s$, 20.5%) (*Figure 3A*, blue trajectories). For P-TEFb, the typical translocation length and the translocation histograms were comparable to those obtained for c-Myc (*Figure 3—figure supplement 2*).

When plotting the MSD as a function of time for c-Myc and P-TEFb, we observed a deviation from linearity for both factors (*Figure 3C*). Such deviation could be due to the 'population exclusion effect' described above ('Materials and methods—Numerical simulations', *Figure 2—figure supplement 2*), but, alternatively, it could also be the signature of an anomalous diffusion process. When a particle undergoes anomalous diffusion, the MSD vs time scales as a power law $t^\alpha$, where $\alpha < 1$ is characteristic of a subdiffusion process (*Saxton, 2007*). However, neither the 'free' Dendra2 nor the c-Myc MSD data could be properly fit by such a law (*Figure 3D*). Similarly to 'free' Dendra2, c-Myc molecules were distributed between populations of very distinct diffusion coefficients. In contrast,

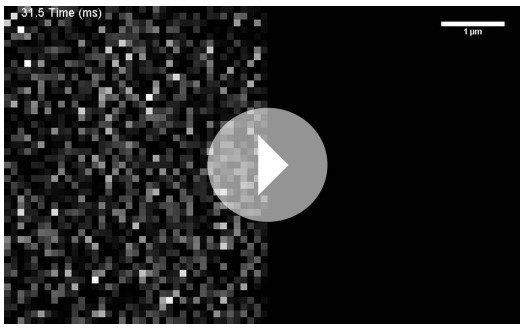

**Video 3**. Raw video of a single c-Myc molecule displaying slow diffusion ($D_2 \approx 0.5$ µm²/s) in the nucleoplasm of a U2OS cell. Running parallel to the raw image, reconstruction of the trace by the localization and tracking algorithms. Exposure time was 10 ms, with 0.5 ms dead time between frames. Running time and scale bars are stamped on the video.

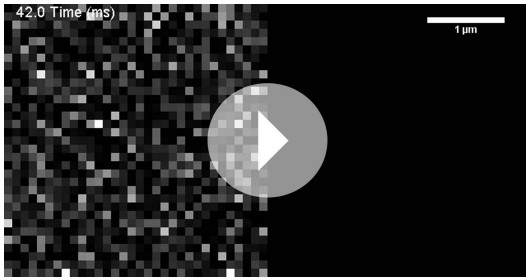

**Video 4**. Raw video of a single c-Myc molecule displaying fast diffusion ($D_1 \approx 13.5$ µm²/s) in the nucleoplasm of a U2OS cell. Running parallel to the raw image, reconstruction of the trace by the localization and tracking algorithms. Exposure time was 10 ms, with 0.5 ms dead time between frames. Running time and scale bars are stamped on the video.

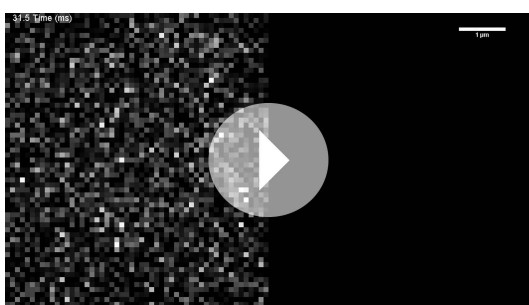

**Video 5**. Raw video of a single P-TEFb molecule diffusing in the nucleoplasm of a U2OS cell. Running parallel to the raw image, reconstruction of the trace by the localization and tracking algorithms. Exposure time was 10 ms, with 0.5 ms dead time between frames. Running time and scale bars are stamped on the video.

for P-TEFb, the MSD variations were remarkably fit by a $t^\alpha$ power law with the anomalous coefficient α = 0.6 (*Figure 3D*). The subdiffusion of P-TEFb was also apparent when we plotted the cumulative histograms of the square displacement for multiples of the time interval (Δt) between two frames and rescaled them by the factor $t^\alpha$, with α determined from the fit in *Figure 3C*. All the rescaled histograms curves collapsed remarkably well for P-TEFb but not for c-Myc or 'free' Dendra2 (*Figure 3—figure supplement 3*). We therefore concluded that the characteristics of single P-TEFb trajectories are consistent with an anomalous diffusive behavior whereas the deviation from linearity of the c-Myc MSD curve reflects the heterogeneity of its diffusion dynamics.

## Asymmetric distribution of angles between consecutive translocations

Subdiffusion in cells is commonly attributed to one of the following two microscopic processes: a broad distribution of trapping times or an obstructed movement resulting from a reduction of the accessible space (*Condamin et al., 2008*) (for a discussion about subdiffusion causes, see 'Materials and methods—Numerical simulations of anomalous diffusion models'). In other words, the subdiffusive behavior, evidenced by the sublinear MSD, is due to either temporal or spatial restrictions. In order to probe the spatial characteristics of the exploration independently of temporal considerations, we analyzed the distribution of angles Θ between two consecutive translocations, an observable that is predominantly sensitive to the geometry of the exploration space (*Liao et al., 2012*) and able to elucidate complex dynamics of molecules (*Burov et al., 2013*).

For 'free' Dendra2 and c-Myc, we found a quasi-uniform angular distribution (*Figure 4A*), as expected for Brownian diffusion. In a three-dimensional space, there is no privileged direction and all angles Θ are equiprobable. In contrast, the angular distribution for P-TEFb was significantly biased toward 180°, reflecting an anti-correlation between two successive displacements. Such anisotropic angular distribution is consistent with diffusion in a space of reduced dimensionality such as a fractal network (*ben-Avraham and Havlin, 2005*). A particle that diffuses in such a structure encounters dead ends, in which case it cannot but return back to previously visited locations (Θ = 180°). Noteworthy, the diffusing subpopulation of H2B molecules also showed a non-uniform angular distribution (*Figure 4—figure supplement 1*).

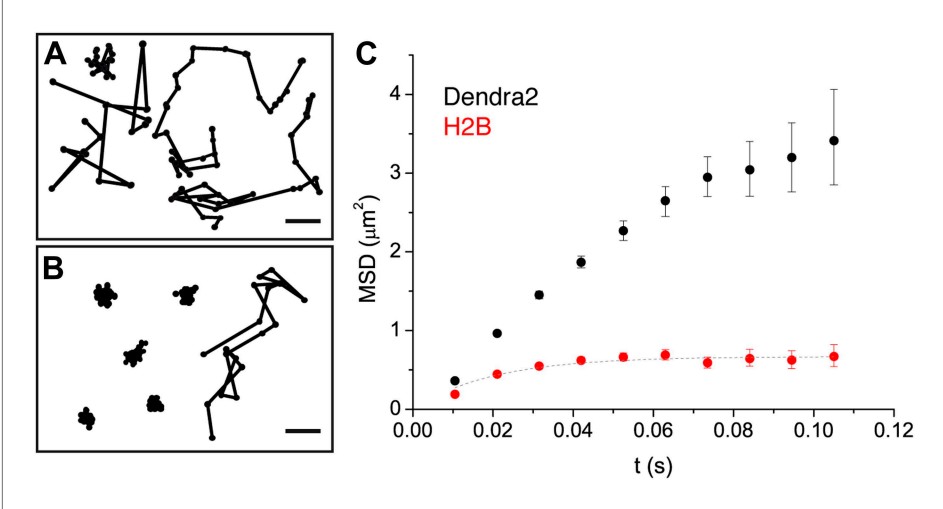

**Figure 2**. Diffusion properties of 'free' Dendra2 and H2B. Examples of single molecule traces of the free fluorophore Dendra2 (**A**) and DNA-associated histone H2B (**B**). In (**C**), the averaged mean square displacement (MSD) as a function of time is represented for both proteins, with an interval of confidence of 95%. The averaged MSD curves were computed from a total of 18,364 trajectories (from 39 cells) for Dendra2, and 40,546 trajectories (from 32 cells) for H2B.

The following figure supplements are available for figure 2:

**Figure supplement 1**. Translocation histograms of Dendra2 and H2B.

**Figure supplement 2**. Population exclusion.

## Temporal evolution and spatial dependence of the angular distribution

An alternative scenario in which such an asymmetric angular distribution of SM traces may arise is that of confined diffusion. If a diffusing particle is confined in a volume of size comparable to the translocation steps, its repetitive bouncing against the trap walls will produce a relative increase of angles larger than 90°. In such a case, the temporal evolution of the anti-persistence reflects the length ratio between the displacement steps and the size of the confining volume. On the other hand, a defining property of fractal structures is their scale invariance (**ben-Avraham and Havlin, 2005**), namely the repetition of structural motifs at different length scales. For a particle diffusing in such a fractal structure, we expected the scale invariance to be apparent in the characteristics of the movement of the particle.

We therefore examined the temporal and spatial dependences of the angular distribution in order to further investigate the origin of the antipersistence of the trajectories and the underlying geometry of the space available for exploration. We defined the asymmetry coefficient (AC) as the logarithm to the base 2 of the ratio between the frequency of forward angles (between 0° and 30°) and backward angles (150°–180°) (**Figure 4B**). The AC is thus negative for angular distributions with a dominant number of backward angles, and it measures the deviation from a homogenous distribution. We calculated the AC for the angles formed at increasing lag times (**Figure 4C**, **Figure 4—figure supplement 1**) as well as a function of the average length of the consecutive translocations forming the angle θ (**Figure 4D**, **Figure 4—figure supplement 1**). It is important to note that with this analysis, the experimental localization accuracy is reflected in the first data point of the spatial dependence of the AC, and not for the data above the 0–150 nm bin. Also, fewer particles are contributing to the AC at larger times, as can be observed in the angular distribution histograms in **Figure 4—figure supplement 1**.

We found out that the angular distribution of c-Myc deviates from homogeneity at increasing lag times with increasing negative AC (**Figure 4C**), potentially reflecting a hindrance to the free diffusion of c-Myc and its confinement to domains significantly smaller than the nucleus. However, the angular distribution became isotropic (AC = 0) at translocations larger than 300 nm (**Figure 4D**). This transition

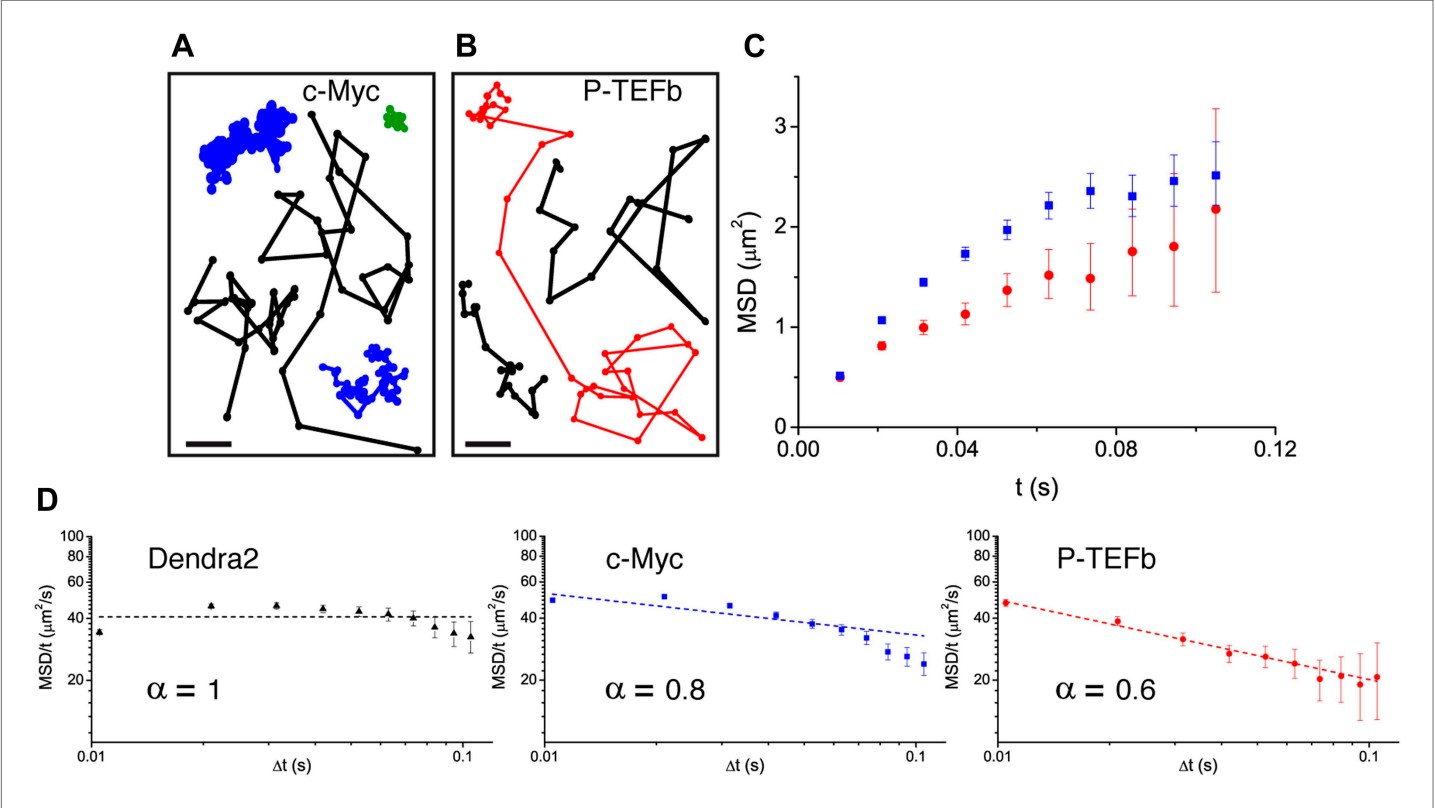

**Figure 3**. Diffusion properties of c-Myc and P-TEFb. For c-Myc (**A**) and P-TEFb (**B**), examples of single molecule traces. From these, we plotted the averaged mean square displacement (MSD) as a function of the lag time with intervals of confidence of 95% (panel **C**), from a total of 33,645 trajectories (from 42 cells) for c-Myc and 16,852 trajectories (from 38 cells) for P-TEFb. In panel **D**, the MSD over time was represented as a function of time in logarithmic scale for 'free' Dendra2, c-Myc and P-TEFb. The fit in the inset follows the time rescaling law MSD(t) = D t$\alpha$, where $\alpha$ = 1 for normal diffusion, and 0 < $\alpha$ < 1 for subdiffusive behavior.

The following figure supplements are available for figure 3:

**Figure supplement 1**. Analysis of the cumulative distribution function of step translocations for c-Myc.

**Figure supplement 2**. Histogram of single translocations for c-Myc and P-TEFb.

**Figure supplement 3**. Cumulative histogram of square displacements rescaled in time.

can be interpreted as an indication of the upper limit size of the confining volume. Hence, the temporal and spatial evolution of the AC suggest two subpopulations of c-Myc, one confined into regions smaller than ~300 nm and a non-confined fraction of c-Myc molecules. P-TEFb, on the other hand, displayed a remarkable constant value of AC for both, time and space (**Figure 4C,D**), possibly reflecting a length-invariant property of the medium in which diffusion takes place.

## Numerical simulations: intermittent diffusion and particle diffusion in media of increasing complexity

In order to gain insight about the different scenarios giving rise to the observed angular distributions, we performed numerical simulations of models with increased levels of complexity (see 'Materials and methods—Numerical simulations' for details about the numerical simulations). In line with the observation of different diffusing populations even for free Dendra2, we first considered an intermittent diffusion model. Here, particles had a probability to switch from a fast to a slow diffusion coefficient and vice versa. We also considered an intermittent trap model, where diffusing particles with fixed diffusion coefficient have a probability to be confined in a spherical trap. We adjusted the parameters of the models in order to obtain similar translocation histograms to those of c-Myc and P-TEFb

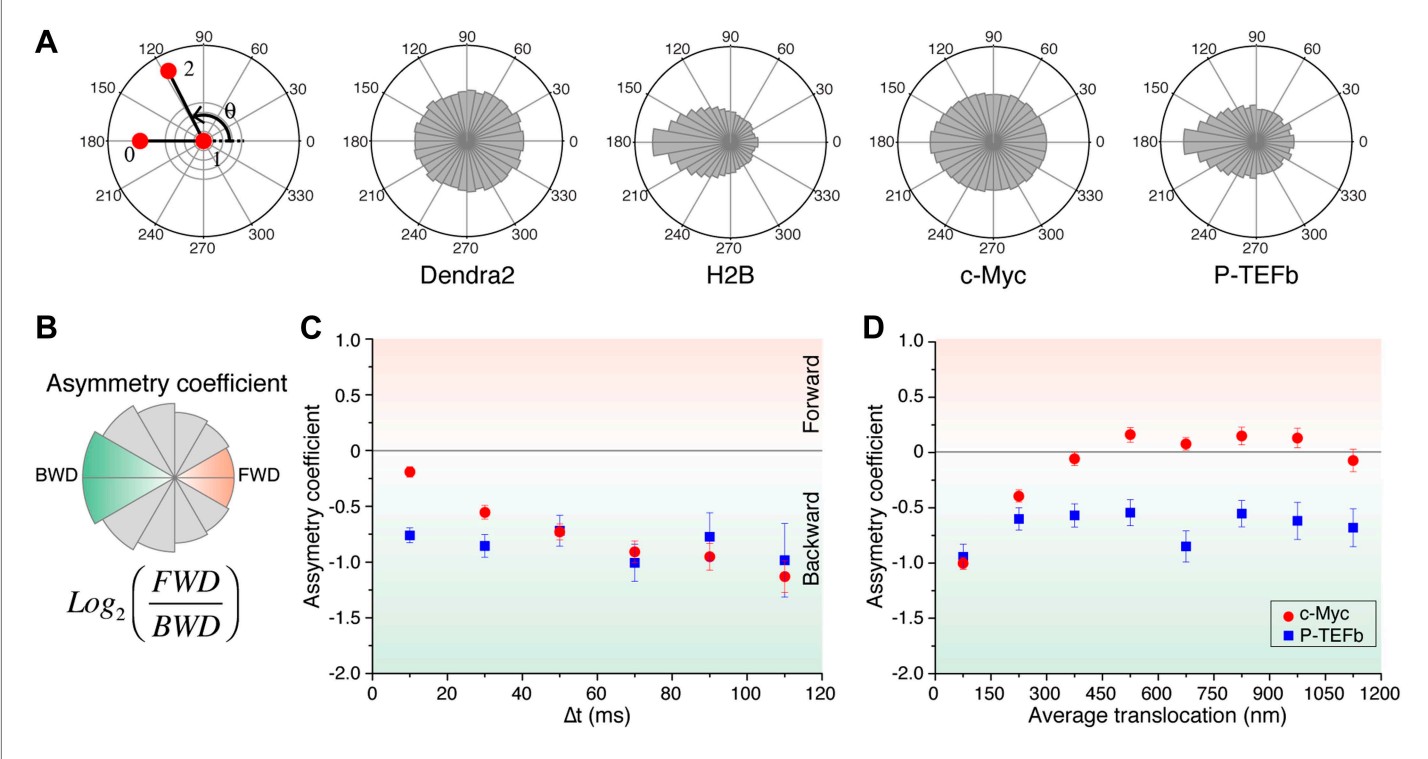

**Figure 4**. Angle distribution between consecutive steps. (**A**) Distribution histograms, in polar coordinates, of the angle θ formed between the vectors of two consecutive translocation steps (vectors formed by positions at time 0 and 10 ms, and between 10 ms and 20 ms), for Dendra2 (23,883 total number of angles), H2B (54,820 angles), c-Myc (46,540 angles), and P-TEFb (13,820 angles). The asymmetry coefficient (AC) was calculated as the logarithm to the base 2 of the ratio between the frequency of forward angles (between 0° and 30°) and the backward angles (150°–180°) (**B**). In panel (**C**), the temporal evolution of AC at increasing lag times has been plotted (i.e., the angle between the vectors formed by the positions at 0 to 10 ms and 10 ms to 20 ms, first data point at 10 ms; angle between the vectors formed at positions 0 to 20 ms and 20 ms to 40 ms, second data point at 20 ms, etc). In (**D**), dependence of the AC with the average translocation value, calculated between the two consecutive steps forming the angle θ and binned at 150 nm. Error bars in (**C**) and (**D**) were calculated as the standard deviation of 50 resamplings using 50% of the data randomly chosen from the radial histograms. Note that the error bars increase as fewer angles are available at increasing lag times and large translocations. Also, how the limited localization accuracy is reflected in the first data point of the spatial dependence of AC in (**D**).

The following figure supplements are available for figure 4:

**Figure supplement 1**. Temporal and spatial dependence of the angular distribution of angles and their asymmetry coefficient (AC).

(**Figure 5—figure supplement 1**, 'Materials and methods—Numerical simulations'). However, none of these simple intermittent models reproduced the antipersistent characteristics of the experimentally measured trajectories (**Figure 5—figure supplement 2**, 'Materials and methods—Numerical simulations').

We then considered a model that results from a combination of intermittent diffusion and intermittent trap. We performed simulations of fast diffusing particles (diffusion coefficient $D_1$) with a probability Kon to engage into a slower diffusion ($D_2$) confined in a trap of radius R (**Figure 5A**). Here, the AC decreased with increasing lag times (**Figure 5B**), reproducing the trend observed in c-Myc. Likewise, the AC displayed the same behavior as c-Myc, tending to zero for larger values of the translocation steps (**Figure 5C**). Following this model, c-Myc performs thus a free exploration of the nuclear space, combined with slower yet still normal diffusion of confined domains, reflecting its interactions with a multiplicity of partners.

Finally, in order to reproduce the invariant properties of the angular asymmetry observed for P-TEFb, we needed to invoke a hierarchical organization of the space. We considered the intermittent trap model, this time with a distribution of trap sizes governed by a Pareto power law (exponent 0.1). With this model, we obtained an antipersistent angular distribution (**Figure 5D**) and

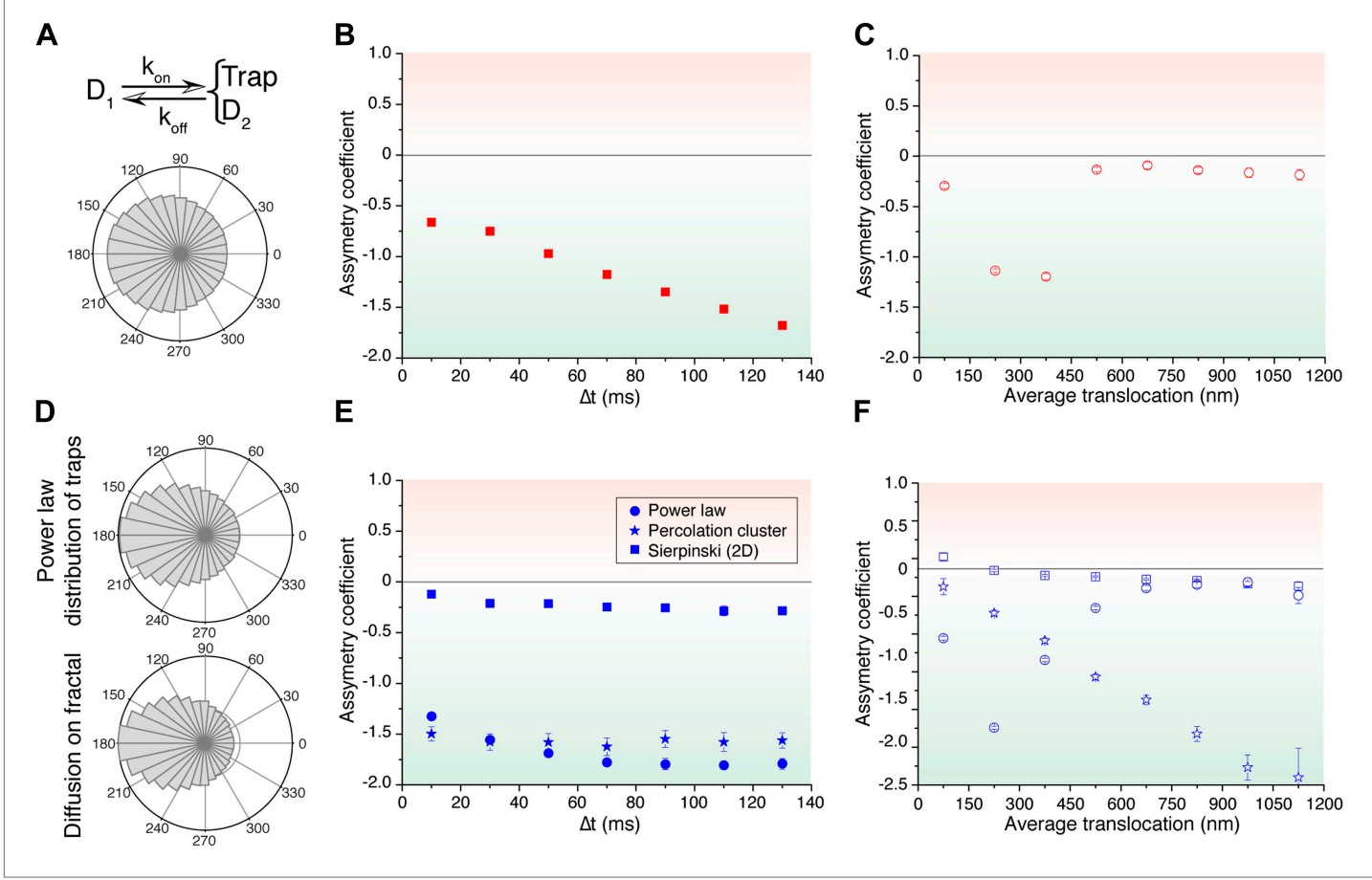

**Figure 5**. Simulated trajectories and distribution of angles. (**A**) Distribution of angles between consecutive translocations for the intermittent diffusion plus confinement model. In this model, a fast diffusing particle with diffusion coefficient $D_1$ has an association rate probability $K_{on}$ to enter into a confined volume (of radius $R_{trap}$) with slower diffusion coefficient $D_2$, and dissociation rate $K_{off}$. (The values of the parameters were $D_1 = 14 \ \mu m^2/s$, $D_2 = 1 \ \mu m^2/s$, $R_{trap} = 500$ nm, $K_{on} = 0.0015$, $K_{off} = 0.02$.). In panels (**B**) and (**C**), the dependence of AC with the lag time and the average translocation step. In (**D**), angular distributions of the intermittent trap simulations with a distribution of trap sizes given by a power law, as well as the simulations of random walks on a percolation cluster. In panel (**E**), temporal dependence of the asymmetry coefficient (AC) for three types of simulations: power law distribution of trap sizes, random walks on a percolation cluster, and random walks on a 2D Sierpinski carpet. In panel (**F**), dependence of the AC with the average translocation size.

The following figure supplements are available for figure 5:

**Figure supplement 1**. Simple models of intermittent diffusion and intermittent confinement.

**Figure supplement 2**. Temporal and spatial dependence of the asymmetry coefficient for the simple intermittent models.

**Figure supplement 3**. Continuous-time random walk.

a closer reproduction of the AC behavior observed for P-TEFb. Although the AC was not strictly constant with time, it did not show a tendency towards zero (*Figure 5E*). Moreover, the spatial dependence of the angular asymmetry formed a plateau for translocations larger than 600 nm (*Figure 5F*).

Such a hierarchy of confining sizes led us to consider a fractal network as an underlying structure on which to simulate the diffusion of particles, also motivated by recent works on the geometry of the nuclear space (*Bancaud et al., 2012*). We considered a 3D percolation cluster as well as a 2D Sierpinski carpet. The Sierpinski carpet is an exact fractal lattice with multi-scale self-similarities. Random walks on a Sierpinski lattice are anomalous because its structure induces spatial correlations between successive displacements. The percolation cluster at the critical percolation threshold possesses the property

of statistical internal self-similarity. As a consequence, the percolation cluster exhibits fractal properties without a defined geometric shape (*ben-Avraham and Havlin, 2005*). For both fractal structures, the angular anisotropy was constant with time (*Figure 5E*), illustrating the scale-invariant features of fractal structures, as observed in the experimental data of P-TEFb. Surprisingly, the AC decreased for larger translocations in the case of the percolation cluster, while the Sierpinski carpet yielded an invariant asymmetry in space. This was interesting because it indicates that the underlying network needs to reproduce a certain degree of geometrical self-similarity, as it is the case of the Sierpinski carpet. The percolation cluster, on the other hand, does not conserve its geometry at different scales but rather other features like the local density obey a power law.

## Target-search and sampling: *compact* vs *non-compact* space exploration c-Myc and P-TEFb adopt opposed search strategies

We have determined that while c-Myc undergoes normal diffusion (with a subpopulation seemingly confined in domains smaller than the nucleus), the dynamics of P-TEFb is well described by a subdiffusive behavior. In the case of P-TEFb, our simulations support the notion that anomalous diffusion is compatible with an obstructed mobility of the proteins, as obtained on a fractal structure (we have ruled out other models of subdiffusion, see *Figure 5—figure supplement 3* and 'Materials and methods— Numerical simulations of anomalous diffusion models' for a more detailed discussion). As previously described, the exponent α = 0.6 of anomalous diffusion obtained for P-TEFb (*Figure 3D*) is a direct measure of the dimension of the walk $D_w = 2/\alpha = 3.3$. Since the fractal dimension $D_f$ has an upper limit at $D_f = 3$, we can therefore conclude that $D_w > D_f$, and thus that P-TEFb is engaged in a *compact* exploration of the nucleoplasm. In contrast, the isotropic sampling of space of c-Myc excludes a *compact* mode of exploration; it undergoes normal 3D diffusion irrespective of its confinement, and hence the dimension of the walk is $D_w = 2$, sampling the nucleoplasm in a *non-compact* manner. These results imply that different factors sense a protein-dependent nuclear environment, which can be determinant for their exploration strategy.

## The distance-dependence of the mean first passage time differs between c-Myc and P-TEFb

The distinctive properties of *compact* and *non-compact* trajectories have potentially important functional consequences on the ability of searchers to find and react with molecular partners. As noted above, a striking difference is the distance-dependence of the mean first passage time (MFPT) of the searcher to the target site. The MFPT of *non-compact* explorers is essentially constant, depending solely on the total volume and not on the distance $r$ to the target. Conversely, in the *compact* case, the MFPT still scales with the volume but also increases with the distance as $r^{(D_w - D_f)}$.

As an illustration, we computed the MFPT as a function of the distance (see analytical expressions of MFPT in *Condamin et al., 2005*; *Bénichou et al., 2010*), using the experimental data for c-Myc and P-TEFb, two examples of *non-compact* and *compact* explorers. For c-Myc, which behaves as an ordinary Brownian walker, the fractal dimension is $D_f = 3$, and the dimension of the walk is $D_w = 2$. We used a diffusion coefficient D = 9.8 μm$^2$/s, the value obtained by a weighted average of the diffusion coefficients of the three subpopulations. (It is important to note that the value used for the diffusion coefficient does not affect the dependence of the MFPT on the initial distance to the target.) To calculate the MFPT, we used a nuclear volume of 600 μm$^3$ and considered a target in its center. For P-TEFb, we did not have direct access to the value of $D_f$ and used several values previously reported as estimations in the nucleoplasm (*Bancaud et al., 2012*). In *Figure 6*, we used $D_f = 2.6$ and the results were qualitatively similar for values of $D_f = 2.2$, and $D_f = 3$ (*Figure 6—figure supplement 1*). For both proteins, we also varied the size $a$ of the target between 1 nm (i.e., corresponding to a couple of base pairs), 10 nm (the size of a protein complex), and 100 nm (the size of a large multimolecular complex).

For c-Myc, the MFPT was constant, irrespective of the distance $r$ (*Figure 6A*). However, it was inversely proportional to the size of the target, similar to what is predicted from the diffusion-limited rate of bimolecular reactions (*Nelson et al., 2008*). In contrast, the MFPT of P-TEFb increased with the distance $r$ but did not depend on the target size. The lack of size dependence can be simply viewed as a consequence of the redundant exploration of *compact* explorers, and reflects the fact that the limiting step to find a target is the time taken to reach its vicinity. We stress that the differences of MFPT can be very significant. For instance, the time needed to find a 10 nm target located at a

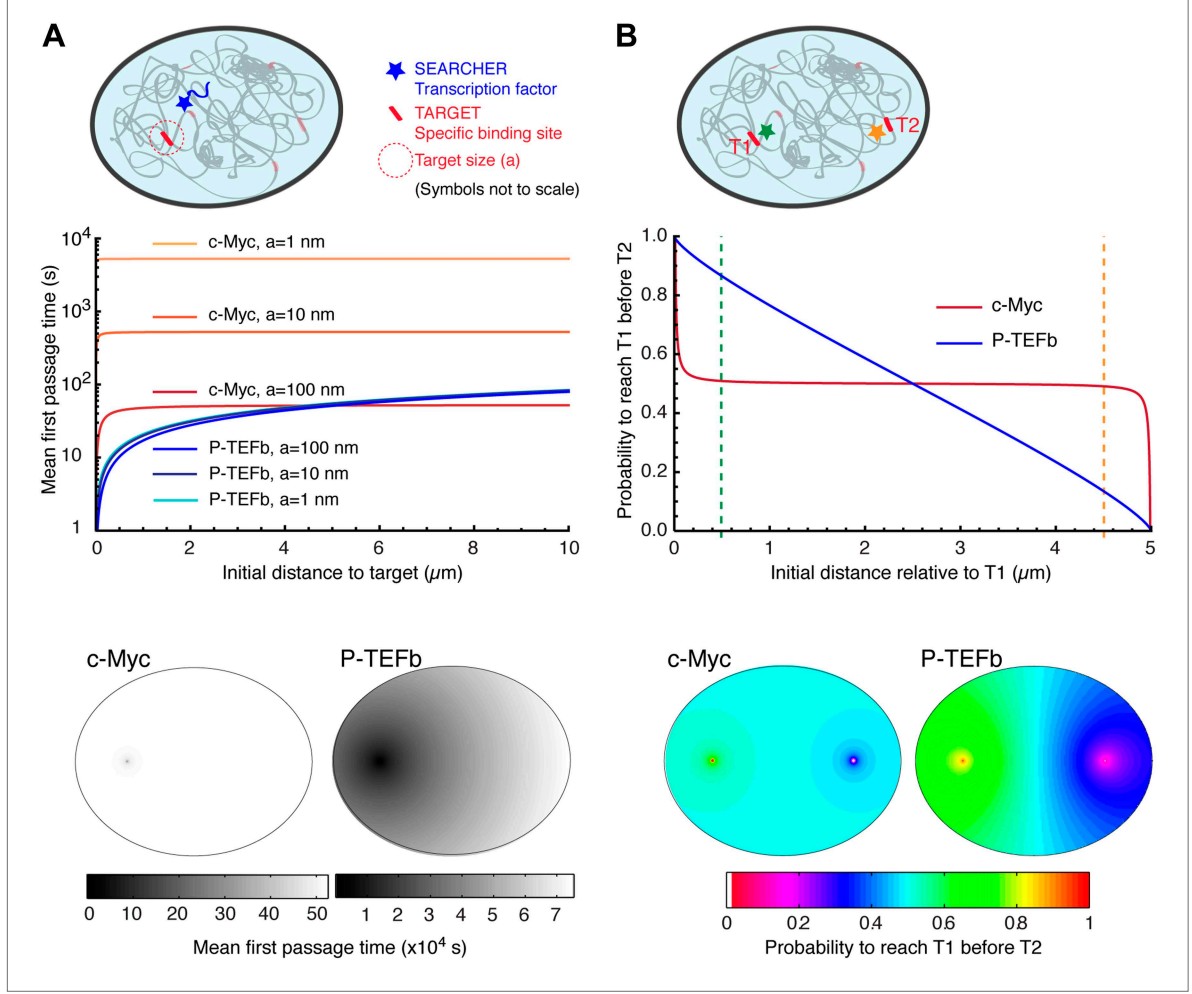

**Figure 6**. *Compact* vs *non-compact* exploration. (**A**) Mean first passage time (MFPT) as a function of the initial distance to the target for both c-Myc (*non-compact* exploration; $D_f = 3$, $D_w = 2$, and diffusion coefficient $D = 9.8$ μm²/s) and P-TEFb (*compact* exploration; $D_f = 2.6$, $D_w = 3.3$, and scale factor of the MSD fit $D = 7.8$). The MFPT was calculated for three different target sizes: 1 nm, 10 nm, and 100 nm. Also, two-dimensional representation of the plots for a = 100 nm are depicted in the lower part of the panel. (**B**) Probability of interaction with target 1 before interacting with target 2, placed at a distance of 20 μm from each other, as a function to the relative distance between the searcher and the targets; two-dimensional plots in the lower side of the panel.

The following figure supplements are available for figure 6:

**Figure supplement 1**. Mean first passage times with $D_f = 2.2$ and $D_f = 3$.

distance of 250 nm is 68 times longer for c-Myc compared to P-TEFb (506.1 s for c-Myc and 7.4 s for P-TEFb). If the target is located at 5 μm of the TF, the difference in the search time is reduced to a factor of 8 (525.3 s for c-Myc and 64.6 for P-TEFb).

Here, we considered that c-Myc has a full access to the nuclear volume. It is interesting to note that if, as suggested by the temporal variance of the angular distribution, c-Myc is confined to a smaller domain, the MFPT would scale linearly with this volume.

We also considered the case of a factor susceptible to bind to two different targets T1 and T2 (***Figure 6B***). To do so, we computed the splitting probability P, that is the probability to reach T1 before T2 as a function of the initial distance to T1. For c-Myc, the probability was equal to 0.5 as soon as the initial distance was larger than a few tens of nanometers, in stark contrast with the case of P-TEFb, for which P varied almost linearly with the distance.

Overall, our analysis of SM experiments of c-Myc and P-TEFb reveals two characteristics of TFs diffusion relevant to the understanding of transcription regulation kinetics. First, the exploration

geometry of the nucleus by TFs is determined by the function and interactions of the nuclear factor. Rather than being subjected to a universal sampling geometry imposed by the nuclear architecture, c-Myc and P-TEFb adopt different modes of exploration leading to normal and anomalous diffusion, respectively. Second, despite apparently similar diffusion coefficients, the different exploration strategies of c-Myc and P-TEFb (*non-compact* and *compact*, respectively) can lead to opposite dependence of the search kinetics on the distance to the target and on the target size. The distance-dependence of the MFPT has direct implications on the probability of interaction of c-Myc and the P-TEFb with their respective partners, which in turn may affect transcriptional kinetics and regulation.

## Discussion

### Protein-specific sensing of nuclear organization

With the PALM imaging assay adapted for SM detection of intracellular proteins in eukaryotic cells, we probed the spatial dynamics of different proteins in the nucleus of live human cells: 'free' Dendra2, histone H2B, the proto-oncogene c-Myc, and the elongation factor P-TEFb. The analysis of individual trajectories, supported by numerical simulations of diffusive tracers on free, confined, and fractal structures, and switching between different regimes, shows that these nuclear proteins fundamentally differ in their exploration of the nucleoplasm. Our results on 'free' Dendra2 are along the lines of those obtained with microinjected fluorescent streptavidin, which explores all nuclear compartments with three subpopulations having different diffusion characteristics (0.15, 0.8, and 5 $\mu m^2$/s) (*Grünwald et al., 2008*), possibly reflecting differences in viscosity and/or crowding in the nucleus. In contrast, FCS experiments using 'free' GFP-repeats or SPT tracking of QD aggregates suggested anomalous diffusion (*Bancaud et al., 2009*).

We determined that 'free' Dendra2 and the proto-oncogene c-Myc undergo normal diffusion in 3D, whereas the displacement of P-TEFb was accounted for by a subdiffusive movement. This finding was further supported by measurements of the distribution of angles between consecutive translocations. Importantly, this distribution was initially isotropic for Dendra2 and c-Myc and an asymmetry towards the return angles increased over time, as expected for confined Brownian motion. Conversely, P-TEFb showed a pronounced and time-invariant anisotropy consistent with the motion on a fractal structure. Thus, the nuclear geometry, or equivalently, the architecture of the space sampled by diffusing factors, is not unique but constitutes a protein-specific parameter. Furthermore, taking into consideration the diffusion parameters derived from the analysis of the MSD, together with the geometrical aspects of the exploration of c-Myc and P-TEFb, we determined the mode of exploration of these factors to be *non-compact* and *compact*, respectively.

We stress that the distinction between *compact* and *non-compact* exploration, rather than the one between anomalous and normal diffusion, is the proper criterion to analyze the search dynamics of transcription factors. The notion of compactness is intimately linked to the geometry and the dimensionality of the sampled space. In this regard, there is a specificity of random motions in a three-dimensional medium with respect to the one- and bi-dimensional cases, for which the exploration is always *compact* since the fractal dimension $D_f$ (less or equal to 1 and 2, respectively) is necessarily smaller than $D_w$. Only in the case of 3D search, can both *compact* and *non-compact* behaviors be observed. Our data demonstrate the relevance of the notion of compactness for the description of nuclear factor dynamics.

### Possible mechanisms controlling the geometry of nuclear explorations

One microscopic mechanism leading to a *compact* exploration of the nucleus could be a compartmentalization of the nucleoplasm into interconnected domains forming a fractal labyrinth in which molecules diffuse. In our view, such a model assuming that molecules encounter physical barriers is poorly compatible with the dynamic nature of nuclear organization and with the lack of correlation between protein size and mobility in the nucleus (*Sprague et al., 2004*; *Mueller et al., 2008*).

Another interpretation is that of a fractal structure restricting the mobility of proteins at its surface. Chromatin has been described as a fractal globule (*Grosberg et al., 2007*; *Lieberman-Aiden et al., 2009*) and transient, non-specific interactions to a continuum of binding sites would account for the diffusing factors not escaping from their interaction with chromatin. In this scenario, the number of binding sites with which c-Myc interacts is not sufficient to restrict its motion to chromatin (36,000 E-boxes in a diploid genome, representing less than 50 sites per $\mu m^3$).

P-TEFb interacts with the CTD of the catalytic subunit of RNA Polymerase II, which contains 52 repetitions of a hepta-peptide motif (*Taube et al., 2002*). The RNA Polymerase CTD is not folded and can occupy the space very efficiently, potentially forming a mesh offering a nuclear continuum of binding sites for P-TEFb. Such CTD matrix could have an intrinsic existence or be linked to the chromatin globular organization. The existence of a nuclear protein scaffold or matrix has been speculated for more than half a century (*Pederson, 2000*) and both our works offer an observation of a functional role for such a structure. Several other studies support this hypothesis, showing that nuclear proteins are in constant interaction with their environment and their motion is governed by specific and non-specific bindings (*Phair et al., 2004*; *Sprague et al., 2004*; *Hager et al., 2009*; *Speil et al., 2011*), therefore opening the door for mechanisms where factors are guided on networks of binding sites (*Bénichou et al., 2011*).

## The effect of the exploration strategy on gene regulation by transcription factors

From a general standpoint, the distance-dependence of the search kinetics could have strong implications for gene regulation. For example, it has been recently shown that, in *Escherichia coli*, the spatial distribution of TFs is determined by the local state of DNA (*Kuhlman and Cox, 2012*). Let us consider the case of TFs co-regulating multiple loci; the relative localization of these loci is an important parameter that will play different roles depending on the *compact* or *non-compact* exploration of the TFs. *Non-compact* TFs have a very similar probability to bind to all loci. In other words, all loci will have the same probability to be occupied, regardless of their spatial position. In contrast, *compact* factors will be preferentially shared between proximal loci, and therefore the probability of a locus to be occupied by a *compact* explorer is a function of the occupation history of its neighboring sites: it is distance and time dependent. Importantly, this indicates that two loci, such as two regulatory sites located a few tens of kbp away from each other, can transfer information and influence one another without direct physical contact. This spatial relation could underlie the process of sequestration of factors away from their targets (*Yao et al., 2011*), which would occur only with *compact* explorers. Such geometrically controlled long-distance interactions are not detectable using conventional chromatin capture assays, which predominantly rely on the chemical crosslinking between contacting sites.

## *Compact* transcription factors and the stability of molecular complexes

A remarkable feature of *compact* searchers is their propensity to visit their neighboring sites multiple times. As a result, they have a probability equal to one to return to a site that they previously occupied, a property designated as the recurrence of *compact* trajectories. From a biochemical viewpoint, this property might affect our understanding of the kinetic stability of molecular complexes. Certainly, molecular machines controlling the nuclear functions such as transcription, splicing, and replication are composed of large numbers of molecules. Some of these molecules are stable constituents while others can be rapidly exchanged in order to control the specificity and modulate the activity of a particular complex (*Fong et al., 2012*). It is therefore important to understand how these molecular machines can assemble from their principal components. For instance, we cannot yet reconcile the need for strong and stable interactions, believed to be required for the viability of such complexes, and the requisite of weak and transient interactions required for molecules to compete for the same target regulating their composition. The observation of *compact* modes suggests that strong binding, associated to small dissociation rates, is not required to ensure high occupancy.

## *Compact* transcription factors favor transcriptional bursting

Recently, the role and importance of transcriptional fluctuations within a single cell have been extensively studied (*Raj et al., 2008*; *Zenklusen et al., 2008*; *Larson et al., 2009*; *English et al., 2011*; *Itzkovitz and van Oudenaarden, 2011*). Using a simple model in which the activation of a gene is controlled by the binding of a single TF to a locus, *Meyer et al. (2012)* have modeled how the search dynamics of these TFs affects the transcriptional response. In this model, for *compact* TFs, the recurrence of the trajectories and the facilitated re-association to the locus would result in transcriptional bursting. In contrast, for the *non-compact* case, the gene activation rate is determined by the total TF concentration in the nucleus, and the transcriptional activity is uncorrelated in time. This further illustrates how the translocation properties of nuclear factors might underlie the kinetics of functional cellular events.

## Conclusion

In this study, we have experimentally demonstrated that different nuclear proteins with different functions sample the nucleoplasm with different search strategies: the exploration geometry of the nucleus is protein-dependent. We have also determined that two different universality classes of search modes, namely *compact* and *non-compact* explorations, coexist in the nucleoplasm. Our current view of the nucleoplasm and chromatin is that of a structure whose condensation influences its accessibility to transacting factors. Here, we have established that the same target in the nucleoplasm can be visited with different probability and kinetics by different factors depending on how they sample space. In addition to chromatin condensation, the compactness of the exploration itself needs to be taken into account to understand how gene regulation operates.

While the space-sampling mode of a random exploration is either *compact* or *non-compact*, the question to be answered in the future is whether one molecule manifests both types of search, exhibiting transitions between them, and whether different dynamics may still arise within each search mode. If that is the case, it will be of paramount importance to understand the level of regulation of such transitions, as well as the implications for the kinetics of the transcription process. The inverse first passage time is a measure of the reaction rate constant. Therefore, the different interaction kinetics that results from *compact* or *non-compact* explorations has profound implications in the understanding of the dynamic interactions and reactivity rates between TFs and corresponding regulated genes. For a *non-compact* explorer like c-Myc, the interaction rate is that of a homogenous solution, and thus it will bind with equal probability to any target in the nucleoplasmic volume. On the other hand, for a *compact* exploration such as the one of P-TEFb, the recurrent search of the local environment and the distance dependence of the search time translate into spatial and temporal correlations between binding events. Such spatial correlation can be seen as a mechanism that adds a level of control to the rapid assembly of molecular complexes, reconciling weak and transient interactions with functional stability. This last notion suggests the idea of a regulated level of compactness of TFs both in time and space.

# Materials and methods

## Cell culture and transfection

U2OS (Human Osteosarcoma) cells were grown in DMEM (Life Technologies, Carlsbad, CA) with 1 g/l glucose and glutamax supplemented with 10% FBS (Fetal Bovine Serum, Life Technologies) and 1% Penicillin/Streptomycin (Life Technologies) at 37°C with 5% $CO_2$. 48 hr prior to the imaging, cells were seeded at 30–40% confluence on a plasma-cleaned (2 min with air with Femto model, Diener Electronic, Ebhausen, Germany) and collagen-coated (Collagen I from Rat tail, Life Technologies) coverslips (N°1 25 mm, Marienfeld, Lauda-Königshofen, Germany).

The C terminal of c-Myc and H2B were fused to Dendra2 and expressed under the control of the CMV promoter. Prior to experiments, U2OS cells were transfected 24 hr before imaging with the plasmid of interest (100 ng/25 mm coverslip) using Fugene 6 (Roche Applied Science, Penzberg, Germany) according to manufacturer's instructions. Clones with very low over-expression of exogenous protein, as judged by low fluorescence intensity of pre-converted Dendra2, were used.

Experiments with P-TEFb (Cyclin T1 fused to Dendra2 on N terminal) and Dendra2 (alone) were performed on U2OS cell line stably transfected and selected with geneticin (Life Technologies). Clones with very low expression of fluorescent protein (CyclinT1-Dendra2 or Dendra2), as judged by low fluorescence intensity of pre-converted Dendra2, were used. Transient transfections of Cyclin T1 Dendra2 were also performed and gave the same results.

## Single-molecule imaging

Single-molecule imaging was performed on an inverted microscope Nikon Ti Eclipse (Nikon Instruments, Tokyo, Japan), with a high numerical aperture objective (1.49 NA) and 100X magnification; extra magnification of 1.5X was used in the tube lens of the microscope, resulting in a total magnification of 150X. We also used perfect focus system (Nikon) designed to avoid drift on the Z-axis (focus) of the objective, relative to the coverslip. The excitation (561 nm) and activation (405 nm) laser beams were injected into a fiber and focused in the back focal plane of the objective, using an appropriate dichroic (Di01-R561-25x36) (*Figure 1—figure supplement 1A*). A motorized mirror allowed us to choose between wide-field or inclined excitation configurations; a small angle, between 0 and 30°,

was typically used to avoid stray-light reflections and reduce background from cell auto-fluorescence. Experiments were acquired under continuous excitation (561 nm laser, 5 kW/cm$^2$ on the sample) and pulsed activation (405 nm laser, 1 pulse of 10 ms per second, 0.01 kW/cm$^2$ during the pulse on the sample). Fluorescence emission from individual Dendra2 molecules was filtered with a single band emission filter centered at 617 nm and a bandpass of 73 nm and recorded on an EMCCD camera (iXon 897 Andor Technology, Belfast, Ireland). The pixel size of the EMCCD was 16 µm, and we imaged a small region of interest (ROI) of about 100 pixels × 100 pixels. This ROI was sufficient for imaging a large cross-section within the nucleus of single cells, and allowed acquisition rates as fast as 100 Hz (10 ms per frame). Images of the pre-converted (green) form of the ensemble fluorescence of Dendra2 were taken using a mercury lamp for illumination (excitation: 485 nm, emission FF01-525/30).

Cells were imaged in Leibovitz's L15 medium (Life Technologies) containing 10% FBS (Fetal Bovine Serum, Life Technologies). The sample was placed on the microscope, on a stage heated at 37°C on the microscope. Once an ROI was selected from the pre-converted (Dendra2 green-form) fluorescence imaging of the live cells, activation pulses were fired every 100 frames, and videos of several thousands of frames were acquired under continuous 561 nm illumination (typically 2000 to 10,000 frames per cell). Each coverslip was used for a maximum of 45 min after placing them on the scope.

## Pre-converted Dendra2 imaging

The same conditions that were used for SM imaging were used to obtain the images of the pre-converted ensemble fluorescence of Dendra2, but exchanging the light source for a mercury Lamp (Intensilight, Nikon) and appropriate excitation and emission filters (485/20 nm and 525/30 nm, respectively). In order to compensate the very weak expression levels, images were reconstituted averaging 100 images of a temporal sequence therefore minimizing the noise.

## mRNA expression and c-myc expression amplification analysis

Based on RNA Pol II chIP-SEQ data available in the laboratory, we selected genes that are expressed in U2OS cells. Those genes were: SPG21, LMF1, BEX1, IGF2R, GAPDH, HMGB2, SOD1, RPL30, ORC3, CUL1, TRAF5, STX11. Using RT-qPCR, we compared the mRNA expression of these genes in two conditions: wild type untransfected U2OS and c-Myc-Dendra2 transfected U2OS. In order to precisely compare the amount of RNA, we counted and fluorescence-activated cell sorted (FACS) the same number of untransfected and c-MYC-Dendra2-expressing cells. We then performed quantitative PCR experiments and compared the expression levels of the analyzed RNA in the two different conditions.

RNA was purified using TRIzol Reagent (15596-018; Invitrogen, Life Technologies) according to the manufacturer's instructions. Total RNA was quantified on a NanoDrop 2000c Spectrophotometer (Thermo Fisher Scientific, Waltham, MA) and their quality was evaluated on RNA Nano Chips (5067-1511; Agilent 2100 bioanalyzer, Agilent, Santa Clara, CA). Reverse transcription from total RNA to cDNA was done with oligo-dT (18418-020; Invitrogen, Life technologies) using SuperScript III RT (18080-085; Invitrogen, Life technologies) and RNAse OUT (10777-019; Invitrogen, Life technologies).

Quantitative real-time PCR (qPCR) was done using 5 µl of 1:20 diluted cDNA on a LightCycler480 system (Roche, Basel, Switzerland) using Maxima SYBR Green qPCR Master Mix (K0252; Fermentas, Thermo Fisher Scientific). A final concentration of 500 nM of primer pairs (Eurofins, MWG, Huntsville, Al, designed according to *Dugast-Darzacq and Grange, 2009*) was used for each qPCR reaction. The cycling conditions were as follows: 95°C for 10 min, 45 cycles (95°C, 15 s; 58°C, 30 s; 72°C, 20 s) and melting curve analysis. LightCycler 480 SW 1.5 was used to evaluate and to analyze the data.

## Analytical methods

### Detection and tracking of single molecules

#### Detection

The diffusion of the molecules imaged in the nucleus of eukaryotes can be as fast as ~10 µm$^2$/s. This implies that the detected molecules can travel a distance larger than the diffraction limit of light (~250 nm) during the characteristic acquisition time (10 ms). The intensity profile is therefore a convolution between the Airy pattern of the point spread function (PSF) and the trajectory of the particle during the 10 ms acquisition time (*Figure 1—figure supplement 1B*). While such motion blur contains potentially useful information (*Elf et al., 2007*), it has some detrimental consequences: a decrease of the SNR and the ineffectiveness of traditional Gaussian fit localization algorithms. An approach,

demonstrated in bacteria, to minimize the blurring effect consists in illuminating the sample with brief (1 ms or less) and intense (up to 100 kW/cm²) laser pulses (**Elf et al., 2007**). In bacteria, this stroboscopic method is all the more necessary since the extension of the motion blur is often comparable to the size of the cell itself (section of 1 µm²). However, given the larger dimension (section ~200 µm²) of a mammalian nucleus, this method requires high laser power, not practical with standard microscopes or live cell microscopy due to phototoxicity effects. Hence, we favored an approach using lower intensity (~4 kW/cm²) and longer illumination time (~10 ms), limited by the readout rate of our camera. At this time scale, trajectories are not affected by the nuclear confinement.

In this case, the emission of fast diffusing single fluorophores cannot be detected with traditional two-dimensional Gaussian fit algorithms (**Cheezum et al., 2001**; **Abraham et al., 2009**). We developed an alternative, comprehensive algorithm capable of detecting fast diffusing molecules that typically have low signal-to-noise ratio (SNR) as well as immobile particles with higher SNR.

For each frame, background intensity was estimated at each pixel as the median intensity of the pixel over the entire video. This background was subsequently subtracted from the raw image. Fluorescence signal from individual molecules may still appear as an aggregate of disconnected pixels, therefore a smoothing step was applied using a Gaussian mask with standard deviation of $\sigma$ = 121 nm. Those pixels with an intensity corresponding to the highest 20% of the non-smoothed (but background corrected) image were selected (**Figure 1—figure supplement 1C**). At such threshold level random noise fluctuations were still included in the pixel selection, we therefore disregarded any spot that spanned less than 0.2 µm² (~20 pixels, or about half the theoretical optical response of the system). Individual pixel aggregates were then selected for each frame, with one additional constraint to account for molecules diffusing outside and back inside the focal plane during the acquisition time; we considered detected spots closer than 500 nm as originating from the same molecule. The position of each spot was calculated as the center of mass of the pixel aggregate, which is a good estimator of the particle position suggested by deconvolution approaches (**Michalet, 2010**).

We tested our detection algorithm with simulated videos consisting of white noise (without single particles signals) with pixel intensity values and standard deviation comparable to the background noise of our live cell data, resulting on a detection rate of $10^{-6}$ detections per frame per µm², three orders of magnitude lower than the typical detection rates obtained with the experimental data.

The localization accuracy of the detection algorithm could also be estimated. We calculated the standard deviation of the position coordinates of a H2B molecule, detected in 290 consecutive frames (3 s tracking) with no apparent diffusion. As shown in **Figure 1—figure supplement 1D**, we obtained a localization accuracy of ~70 nm.

## Tracking

In order to connect consecutive detections of one given molecule, we defined the maximum distance $R$ allowed for the translocation of a single step of the particle. For each single particle detection, the radius $R$ defined an area centered on the particle position at time $T$ on the consecutive frame at $T + \Delta t$. When a detection at $T + \Delta t$ was found within the area defined by $R$, the two detections were linked in a trajectory. When two or more particles were detected within this area, the trajectory was truncated and the positions considered as the first detection of new trajectories. When the detection of one particle could be included in two different trajectories, both trajectories were also truncated, and the detection was disregarded. We defined such a restrictive policy of tracking in order to reduce the number of misconnections, or false-positive tracking connections. A misconnection occurs when two consecutive detections from two different molecules are included in the same tracking sequence. Therefore, when there is any ambiguity between two spatially closed detections, the algorithm truncates the trajectories.

Such restricting tracking policy reduced the number of misconnection but also reduced the total number of traces suitable for analysis. Therefore, in order to set an appropriate maximum radius $R$, we computed the probability of detecting two different molecules in consecutive frames within a distance lower than $R$.

## Maximum tracking radius $R$ and misconnection probability

Considering the detection of a given molecule in consecutive frames, we could estimate the probability of tracking error by determining the local density of molecules different than the molecule of interest. In order to do so, we determined the local particle density at a time point where the probability of detecting the same particle is close to zero. We estimated the fluorescence photobleaching

characteristic time under our experimental conditions by measuring the fluorescence lifetime of an ensemble of proteins in the nucleus, after a high intensity activation pulse, and under usual imaging conditions (*Figure 1—figure supplement 2A*). We measured a fluorescence half-life of ~600 ms, suggesting that the probability of a molecule photobleaching between two consecutive frames is 0.02. After 5 s (476 frames) of the initial detection, it is highly improbable (0.0001 probability) that a detection originates from the same molecule. For each molecule detected, we could therefore calculate the number of detections around the same spatial coordinates but at a time separation of 5 s or more and thus estimate the average local density within a radius $R$ of the molecule.

Considering the set of all the detections $(x_i, y_i, t_i)$, where $(x_i, y_i)$ are the spatial coordinates and $t_i$ the time, for each detection $i$, we defined $N_i$, the total number of frames recorded more than 5 s after each given detection $i$. The detections made during this period are estimators of the local density around detection $i$. We therefore defined $W_i(R)$ as the total number of detections during these time-shifted frames within a distance smaller than $R$ (i.e., the total number of detections within $R$, after 5 s). Being $M$ the total number of detections, we could calculate the total number of expected misconnections within a radius $R$ as follows:

$$W(R) = \sum_{i=1}^{M} \frac{W_i(R)}{2N_i}$$

By comparing the value obtained from this expression to the number of connections we measured, we could estimate the probability to misconnect two detections. The total number of connections $C(R)$ could be then calculated integrating both, the misconnections and the positive translocations. For every detection $i$ we calculated $C_i(R)$, the number of detections at the consecutive frame at a distance smaller than $R$. The sum of all detections was therefore:

$$C(R) = \sum_{i=1}^{M} C_i(R)$$

In *Figure 1—figure supplement 2B*, we plotted the measured $W(R)$ and $C(R)$ as well as their difference, for the free fluorophore Dendra2 as well as for all the proteins under study. We observed that for $R$ bigger than ~2 µm the total number of tracking assignments was dominated by misconnections; we therefore set the maximum allowed radius $R$ for tracking under our imaging conditions to be 2 µm.

A tracking misconnection could occur at the first or last translocations of a trace, or in the middle of the trajectory. If a false connection occurred in the middle of the trace, its origin was the erroneous link of two traces from different molecules, being the first one detected until frame $i$ and the second one starting at frame $i + 1$, appearing in the vicinity of the first molecule. The probability of such event to happen is very low because the number of single frame detections outnumbered by two orders of magnitude the number of trajectories with at least two consecutive detections. We could therefore consider that the tracking error misconnections occurred mainly at the beginning or end of the trajectories. Taking into account only the detections that are at the extremities of a trajectory, after tracking with a maximum radius of 2 µm, we could therefore estimate the probability of false connection from the fraction of misconnections as:

$$P^W = \begin{cases} 0 & \text{Within trajectories} \\ P^w_{extremity}(d_i) & \text{for the first and last steps} \end{cases} \quad \text{and} \quad P^w_{extremity}(x) = \frac{W'(x)}{C'(x)}$$

where $W'$ and $C'$ stand for the first order derivative of $W$ and $C$, which were estimated for mathematical convenience every 0.05 µm and then linearly extrapolated.

To extend the notion of one-step translocation error probability to several steps displacement, the probability for a trajectory to be false was set to be the mean of all the one-step translocations that composed the trajectory.

## Cumulative histogram analysis and mean square displacement

The data of proteins diffusing in the nuclear volume is the 2D projection of a 3D motion. Provided that the nucleus is isotropic along the three spatial axes X, Y, and Z, the XY projection data fully reflect the 3D behavior of the molecules.

The analysis of the cumulative translocations histogram allows for the determination of individual components from a mixed set of translocations, that is, translocation steps that cannot be governed by a single diffusion coefficient. For the 1Δt time step (10.5 ms), the cumulative distribution function (CDF) is a function $F_1(x)$ that represents the probability that a random translocation may be found at a distance smaller than $x$.

The cumulative function weighted with the probability of misconnection is:

$$\overline{F_1}(x) = \frac{\sum_i H(x - d_i)(1 - P^w(d_i))}{\sum_i (1 - P^w(d_i))}$$

where the sum is computed for all the recorded translocations $d_i$, $H$ represents the Heaviside step function, which is 1 for $x - d_i \geq 0$ and 0 for $x - d_i < 0$ and $P^w$ is the misconnection probability described in the previous section.

The probability distribution of 2D translocations for a single diffusion coefficient $D$ and an inter-frame lag time $T$ is given by $\frac{1}{\sqrt{4DT}}e^{-\frac{x^2}{4DT}}$. This imposes a CDF of translocations for a set of translocations that can be described by single population of diffusion to be:

$$1 - e^{-\frac{x^2}{4DT}}$$

When the measured translocations reflects a pool of molecules with different diffusion kinetics, this single exponential function fails to describe the empirical CDF. In the case of k diffusing species, the empirical CDF is best described by

$$1 - a_1 e^{-\frac{x^2}{4D_1 T}} - a_2 e^{-\frac{x^2}{4D_2 T}} - \cdots - a_k e^{-\frac{x^2}{4D_k T}}$$

where $a_i$ represents the fraction of translocations in the probability distribution imposed by a diffusion coefficient $D_i$. The normalization condition $a_1 + a_2 + \cdots + a_k = 1$ has to be satisfied, and $k$, the number of different diffusing populations, is as small as possible.

The evaluation of the CDF for different time lags (Δt = 1, 2, … 10) allowed us to estimate the individual diffusion coefficients (*Schütz et al., 1997*) (*Figure 2—figure supplement 1*). This analysis of the cumulative distribution function fits the experimental data with a model of Brownian diffusion of different populations. Further analysis of the mean square displacement of translocations and the step correlation was necessary to determine the nature of diffusion and validate or refuse the simple Brownian model independently for each protein.

## Mean square displacement (MSD)

We first computed the mean square displacement of translocations for each individual trace $j$ ($MSD_j$) of length $n$, weighted with the probability of misconnection previously described. The $MSD_j$ is therefore:

$$MSD_j(t) = \frac{\sum_{i=1}^{n-t} d_{i,i+t}^2 (1 - P^w(d_{i,i+t}))}{\sum_{i=1}^{n-t} (1 - P^w(d_{i,i+t}))}$$

where $d_{i,i+t}$ is the translocation distance between the frames $i$ and $i + t$. The $MSD_j$ for individual traces was then computed for increasing lag times up to *10Δt* ($t = 1Δt, 2Δt, … 10Δt$) where *Δt* is the experimental inter-frame time interval of 10.5 ms. We then calculated the average mean square displacement *MSD(t)* for $t = 1Δt, 2Δt, … 10Δt$ as the mean of all the individual traces $MSD_j(t)$ for all the trajectories that had a length of at least equal to $t$. Finally, error bars for each data point of the average *MSD(t)* were calculated as the 95% interval of confidence computed by bootstrap resampling of the population.

## Numerical simulations

In order to validate our detection and tracking algorithms and analysis, we performed a series of numerical simulations. These simulations consisted in videos of particles with the optical response of

our optical system, diffusing in 3D Brownian motion with a given diffusion coefficient. The signal was then corrupted with additional noise composed by a mixture of Gaussian and shot noise that mimicked our raw experimental data.

## Parameters of the simulation

The PSF of the single particle signal was obtained with the PSF Lab software (*Nasse and Woehl, 2010*). The parameters used to retrieve the PSF were: emission wavelength 600 nm, objective NA 1.49, coverslip thickness 150 μm, oil refractive index 1.51, coverslip refractive index 1.52, and sample refractive index 1.3. The PSF was computed for a total height of 4 μm in layers of 100 nm and radius of 2 μm, on a pixelated image with pixel size of 107 nm. Intermediate values were estimated by linear interpolation of the eight surrounding points.

In order to estimate the noise, we analyzed the distribution of the pixel intensity values of experimental videos after removing the values of those pixels included in any detection. We then fitted this distribution to a combination of Gaussian and Poisson distributions, with a result of 95% Poisson distribution ($\lambda = 20$) multiplied by a factor determined by the camera gain, and 5% white noise.

The movement of the particles was simulated to take place in the interior of a closed box with similar dimensions to those of the eukaryotes nucleus: 10 μm × 10 μm × 6 μm. The particle density inside the box was set to be constant (i.e., the photobleaching rate and the photoactivation rate were the same), and therefore the ratio of disappearance and appearance of a new particle at a random position was set accordingly to the measured photobleaching half-life.

The video images were finally obtained as a convolution of the PSF with the displacement of the particle in the pixelated matrix during the acquisition time. We computed this by estimating the position (x, y, z) every 1 ms (10 estimations per frame) and by adding the convolved PSF at (x, y, z) to the final image. To take into account the displacement during the EMCCD transfer time between two consecutives images, an additional unrecorded movement of 0.5 ms was added to the simulation.

## Reconstruction of diffusion

We then run the simulated videos of one single population of 3D Brownian diffusing particles through our detection and tracking algorithms. The histogram of translocations retrieved from the analysis of simulated films was in very good agreement with the theoretical values for diffusion coefficients between 0.1 μm²/s and 20 μm²/s (*Figure 1—figure supplement 3A and 3B*).

## Diffusion coefficient boundaries

In our experiments, the minimum inter-frame displacement was limited by the experimental single molecule localization accuracy. The pointing error can be defined as the distance between the real centroid of the particle and the coordinates of the detection (*Figure 1—figure supplement 3C*). Using our simulations, we could determine the localization accuracy as the mean value of the pointing error, as a function of the diffusion coefficient. It was estimated to be ~70 nm with a dramatic increase for particles with diffusion coefficient higher than 10 μm²/s. This is in agreement with the experimental estimation of the localization accuracy retrieved from the consecutive detections of an immobile H2B molecule (*Figure 1—figure supplement 1D*). The lower bound of a detectable diffusion coefficient was thus ~0.04 μm²/s.

The analysis of simulated videos also allowed us to determine the percentage of detected particles. We could determine that the percentage of detections followed a Gaussian-like distribution along the optical axis, centered at the focal plane (*Figure 1—figure supplement 3D*). The width of such distribution determined the focal depth and it is in good agreement with the axial width of the PSF in our experimental conditions (~600 nm). Moreover, the amplitude of the detection distribution was dependent on the diffusion coefficient of the particles. There is a higher rate of detection for slow particles than for those with higher diffusion coefficient. This effect could also be observed in the in vivo data by plotting the averaged *1Δt* displacement as a function of the duration of the trajectory (*Figure 2—figure supplement 2A*, for Dendra2). Less mobile particles were detected for longer periods of time.

## Population exclusion

The dependency of the percentage of detected particles with the diffusion coefficient of the particles needs to be taken into account when analyzing the mobility of a heterogeneous mixture of molecules

with different diffusion coefficients. We performed simulations of an extreme case with 50% of the molecules following Brownian diffusion at 1 μm²/s and 50% at 10 μm²/s. As expected, fast particles had a higher probability of escaping the focal depth of observation between two consecutive frames, and therefore slow particles were over-represented in the measurement (*Figure 1—figure supplement 3C*). Such exclusion of the fast diffusing particles population significantly affects population analysis as well as the average MSD. The population analysis of the one step translocation histogram gave a proportion rate of 80% of particles with D = 1 μm²/s and 20% D = 10 μm²/s. Despite the bias on the population, the values of diffusion coefficients were not affected by the population exclusion (*Figure 2—figure supplement 2C*). Similarly, the resulting MSD analysis of the simulated data showed a deviation from linearity, suggesting an apparent subdiffusive behavior of the ensemble of molecules (*Figure 2—figure supplement 2D*).

In order to take this bias into account in our analysis, we measured the number of translocations detected on single particle simulation videos with diffusion coefficients ranging from 0.001 μm²/s to 20 μm²/s (*Figure 2—figure supplement 2B*). We then used this information as the reference curve to correct the proportions of populations retrieved from the analysis of our experimental data.

In order to perform such correction, we considered an arbitrary fit of the one step translocation histogram with three populations: $(a_1, D_1)$, $(a_2, D_2)$ and $(a_3, D_3)$, with $a_1$, $a_2$, and $a_3$ representing the fractions of populations and $D_1$, $D_2$, and $D_3$ their diffusion coefficients. We found, interpolating the reference curve for each diffusion coefficient, the proportion of the population that was integrated in our study p($D_1$), p($D_2$), and p($D_3$). For instance, when D = 1 μm²/s, we detected p(D) = 97% of the translocations. We then computed the corrected values for $a_1$, $a_2$, and $a_3$ as $\frac{a_1}{p(D_1)}$, $\frac{a_2}{p(D_2)}$, and $\frac{a_3}{p(D_3)}$ to obtain the relative rate of diffusive populations.

It is important to note that this correction assumes Brownian diffusion of the molecules, and therefore has to be understood as a first order correction of the population rates in all our experimental data. However, due to this selection bias, fast diffusing molecules, for which the MSD slope is the highest, contribute less to the average MSD at longer time lags. As a result, the average MSD observed for Dendra2 is consistent with a normal diffusive behavior for three species.

## Simulations of models with intermittent regimes

With our experimental system well characterized, we simulated increasingly sophisticated models of diffusion with intermittent regimes. We first considered a model in which particles transition from fast to slow Brownian diffusion. We also considered a model in which particles switch from free diffusion to confined diffusion within a spherical trap. In both cases, we considered an infinite volume in the plane x–y, and to 1 μm in the axial direction. We then recorded the positions of simulated traces until they exited the volume. For each model, we simulated 100,000 trajectories whose recorded position was corrupted by a Gaussian curve with 70 nm standard deviation, simulating the experimental localization accuracy.

We first simulated regime switching between a fast (D1) and a slow (D2) diffusing coefficient. For each translocation, there was a probability Kon to transition from D1 to D2, and a probability Koff to transition from D2 to D1. We adjusted D1 and D2 to the diffusion coefficients obtained from the two-population fit of the experimental cumulative histograms of steps, for both c-Myc and P-TEFb. These values were D1 = 14 μm²/s and D2 = 1 μm²/s for c-Myc, and D1 = 15 μm²/s and D2 = 1 μm²/s for P-TEFb. Likewise, we determined the ratio Kon/Koff from the retrieved populations of the fit (Kon/Koff = 0.68/0.32 for c-Myc, Kon/Koff = 0.54/0.46 for P-TEFb). With these constrains, we found the association and dissociation rates that resulted in a translocation of histograms close to those obtained in the experiments (*Figure 5—figure supplement 1*).

We also investigated switching between a freely diffusing mode and a confined diffusion. The potential of confinement was set to 'hard-wall' type, with random repositioning of the molecule in the volume in the case of two successive bouncing against the boundaries. A new confinement was created anytime the particle switched between a freely diffusing and confined mode.

With the intermittent diffusion model, we retrieved mild negative values of the AC. These were prominently at small lag times and small translocation steps. The tendency for larger times and steps was always toward AC = 0, that is a symmetric angular distribution (*Figure 5—figure supplement 2*). We concluded that such scenario could not reproduce the antipersistent characteristics of our data. Conversely, we obtained stronger angular asymmetries with the intermittent trap model. The AC

curves also evolved with time and space and were strongly dependent of the size of the trap. Expectedly, the simulated particles undergo confined diffusion only when they are inside the trap. In conclusion, although we reproduced certain characteristics of the angular distribution of our data, these simple simulations failed to reproduce the AC temporal and spatial dependences of c-Myc, as well as the scale independent behavior of P-TEFb.

## Numerical simulations of anomalous diffusion models
### Models for subdiffusion

Subdiffusion motion has been frequently reported in SPT experiments (*Saxton, 2007*). In cells, it is commonly attributed to one of the following two processes: a broad distribution of trapping times or an obstructed movement due to crowding effects. In our experiments on P-TEFb, we could rule out the former, often referred to as the continuous time random walk model (CTRW) (*Metzler and Klafter, 2000*). We addressed this model by simulation to compute its angle distribution. In *Figure 5—figure supplement 3*, we show the result of a Monte Carlo simulation of a continuous time random walks performed on an infinite cubic lattice. The position was recorded every 1000 steps, and the waiting times were uncorrelated following a discretized heavy-tailed probability distribution,

$$\begin{cases} P(T < 1) = 0 \\ P(1 < T < t) = 1 - t^{-\alpha} \end{cases}$$

with α set as 0.6. The MSD shown in *Figure 5—figure supplement 3A* is an ensemble MSD, averaged on 10,000 trajectories and rescales as a power law $\langle r^2(t) \rangle \sim t^a$. It is noteworthy to point out that the time-average MSD of a CTRW realization does not result in a sublinear relationship with time (*ben-Avraham and Havlin, 2005*). The experimental MSD curves shown in *Figure 2* and *Figure 3* were averaged over time and also over the ensemble of all the trajectories, which was an additional indication against the CTRW model for our data.

Random walks on a fractal medium induce spatial correlation between successive displacements imposed by the self similarity of the geometry. For that reason, the fractal model has been applied to the comprehension of random walks on disordered media (*Szymanski and Weiss, 2009*). Fractal object involves the repetition of the same features of an object at different scales. If the whole object is repeated then the fractal is exact. One example of exact fractal is the Sierpinski gasket, a 2D-embedded fractal lattice of dimension df = log(8)/log(3) ≈ 1.89. An example of a non-exact fractal is the maximum site percolation cluster at percolation threshold. The geometry is not conserved in the cluster at different scales, but rather features such as the local density obey a power law.

We computed a fractal network as the maximum cluster of a cubic lattice at critical site percolation probability (*ben-Avraham and Havlin, 2005*). The initial cubic lattice dimensions were 2000 × 2000 × 500 sites. We then removed sites from the lattice according to the critical probability $P_c = 0.311604$ (*ben-Avraham and Havlin, 2005*). The size of the maximum cluster was 5,967,870 sites. We then performed random walks on such fractal structure by recording a position every 2000 steps on the lattice.

### Angular distribution evaluation by Monte Carlo simulations

Simulations of trajectories on cubic and fractal lattices were performed in order to obtain the angular distribution of consecutive steps, and their temporal evolution. The angular distribution was obtained by Monte Carlo simulations of 10,000 realizations of trajectories of 500 steps with a randomly distributed start. Since the initial mesh was a cube, there were privileged directions with higher number of sites and thus a higher number of possible successive positions. In computations of the angular distributions on such simulated trajectories, this results in an over representation of the directions imposed by the lattice geometry such a 90°angle that vanished with an increasing time lag. This bias would be negligible if the percolation cluster had a large number of sites, such that allowed us to record the position of the trajectory for at time lag significantly larger than 2000 steps. The limitation to perform such a simulation was the random access memory of the computer. We therefore applied a simple correction to the angular distribution of the simulated trajectories. We recorded all the possible translocations from our simulations for time lags 1Δt to 10Δt. These translocations were then shuffled to compute the distribution of the angles that was inherent to the network itself and not to the successive displacement correlation. Such 'structural distribution' reflected therefore the anisotropy of the

structure due to the finite scale of the computations, and we used it as a normalization distribution. We verified that this 'structural' angular distribution flattened at increasing time lag. The angular distributions shown in *Figure 5* were therefore rescaled by dividing each bin proportion by the corresponding one in the structural distribution.

## Acknowledgements

We would like to thank Dan Larson, John Lis, Florian Mueller, Yitzhak Rabin, Robert Singer, and Robert Tjian for discussions and comments. Leonid Mirny for an in depth review and his challenge and help with the modeling approach. We are grateful to Sarah Moorehead and Mohamed El Beheiry for critical reading and to Daniel Ciepielewski and Philippe Rideau for discussions on the microscopy. VR acknowledges financial support from FRM, and II from NWO. Work presented here was supported by the PCV DYNAFT 08-PCVI-0013 and DynamIC ANR-12-BSV5-0018 from Agence Nationale pour la Recherche, and a research contract with Nikon France to XD and MD.

## Additional information

### Funding

| Funder | Grant reference number | Author |
| --- | --- | --- |
| Agence Nationale de la Recherche (L' Agence Nationale de la Recherche) | PCV DynaFT | Raphaël Voituriez, Olivier Bensaude, Maxime Dahan, Xavier Darzacq |
| Nikon France | | Ignacio Izeddin, Maxime Dahan, Xavier Darzacq |
| Netherlands Organisation for Scientific Research (NWO) | Rubicon | Ignacio Izeddin |
| Fondation pour la Recherche Médicale (Foundation for Medical Research in France) | | Vincent Récamier |

The funders had no role in study design, data collection and interpretation, or the decision to submit the work for publication.

### Author contributions

II, VR, Conception and design, Acquisition of data, Analysis and interpretation of data, Drafting or revising the article; LB, CD-D, Acquisition of data, Drafting or revising the article; IIC, OB, RV, Analysis and interpretation of data, Drafting or revising the article; LB, Acquisition of data, Analysis and interpretation of data; FP, Acquisition of data, Contributed unpublished essential data or reagents; OB, Drafting or revising the article, Contributed unpublished essential data or reagents; MD, XD, Conception and design, Analysis and interpretation of data, Drafting or revising the article

### Author ORCIDs

Ignacio Izeddin, http://orcid.org/0000-0002-8476-3915

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
