## [Decision Letter]

Thank you for sending your work entitled “Distinct target search modes for c-Myc and P-TEFb revealed by single molecule tracking in live cells” for consideration at *eLife.* Your article has been favorably evaluated by a Senior editor, a Reviewing editor, and 3 reviewers, one of whom, Leonid Mirny, has agreed to reveal his identity.

The Reviewing editor and the other reviewers discussed their comments before we reached this decision, and the Reviewing editor has assembled the following comments to help you prepare a revised submission.

All reviewers felt that the work was a significant contribution and was novel in its approach to the problem of target search. Yet the reviewers also felt that the work could be significantly improved in the clarity of the writing and in considering other models. In particular, Reviewer 3 had an alternative explanation for the data that needs to be considered. He states “To summarize, my recommendation to the authors is to revaluate their conclusions concerning anomalous diffusion and, at least, to consider intermittent binding as an alternative mechanism. Suggested analysis and simulation may allow estimating sizes of targets and dwell times for c-Myc and P-TEFb. I believe this analysis can only strengthen this solid and important paper.” We suggest you address the issues raised by this and the other reviewers by using the intermittent binding model in your simulations to determine the fit for the existing data on angular distribution.

Reviewer #1:

The manuscript presents interesting results regarding the random motion of two fluorescently-labeled transcription factors (TFs) in mammalian cells. The authors suggest that the different spatiotemporal dynamics exhibited by the TFs represent radically different ways in which they experience the geometry of the nucleus, and that this “protein-specific geometry of the nucleus” may have important consequences for transcriptional regulation.

The work is of significance to our physical, quantitative understanding of gene regulation. It is multidisciplinary in nature, involving state-of-the-art imaging methods, advance image analysis and insightful incorporation of ideas from condensed matter physics.

On the other hand, I think the manuscript can be significantly improved in terms of narrative flow and data presentation. Specifically:

1) I wish the key concept, of compact versus *non-compact* exploration of space, was introduced earlier and used to actively guide the narrative. As the text is currently written, the reader is exposed to a lot of data that is pretty standard *–* SPT trajectories and their analysis as diffusive motion *–* before he/she encounters the novel aspects of the work. This burying of the lead is against the authors' own interest, I think.

2) While an overabundance of optics and imaging details are provided in the main text, essential biological details are missing. For example, it is not satisfactorily described how the fusion TFs were expressed, and what evidence there is that the fusions behave like the unfused wild type. The authors “tested the functionality of c-Myc-Dendra2 by performing RT-qPCR on a set of active genes in our U2OS cell line”. Was the fusion expressed in the null background? From the endogenous promoter? Does the fusion protein rescue the null? And as for the other fusion, CycT1-Dendra2, we are pointed to a work “in preparation” and are thus unable to judge for ourselves the evidence for its functionality.

3) The presentation of experimental data is also less than optimal. Examples: (a) Sample sizes (# trajectories, # cells, etc.) need to be stated for each plot, but they are not. (b) The evidence for sub-diffusion (Figure 3) is not convincing. The anomalous behavior of MSD should be demonstrated for individual trajectories, not for an ensemble average. (c) The evidence for time-evolution of angle preference (Figure 4) is also unconvincing. A single parameter should be extracted from the plot at each time point, and the change of this parameter value over time should be examined to reveal whether the angle preference changes over time or not. This would allow direct comparison to theory (Figure 5).

Reviewer #2:

This manuscript presents an elegant work of imaging single molecules of transcription factors in the nucleus of living cells by single particle tracking photoactivation localization microscopy (sptPALM). They have performed elegant microscopy and data analysis to collect large data sets of individual proteins diffusing within the nuclear volume. High quality single molecule trajectories were acquired and reconstructed for as long as 60 consecutive translocations. Their work provides an experimental framework to study nuclear proteins with mobility ranging from chromatin-bound H2B molecules to diffusing Dendra2 molecules as a general method for studying nuclear dynamics at the single molecule level.

This manuscript has raised several interesting points on transcription imaging in living cells. The authors describe the concept of compact and *non-compact* exploration of transcription factors. The compact exploration modes suggest that strong binding is not required to ensure high occupancy and that compact exploration factors may be preferentially shared between proximal loci. They also suggest that spatial cooperation can be a mechanism that adds a level of control of rapid assembly of molecular complexes, reconciling weak and transient interactions with functional stability.

Specific points:

1) The actual data presenting the described results (diffusion coefficients, etc) cannot be found.

2) It is interesting that free Dendra2 and c-myc have several populations of very distinct diffusion coefficients. Stokes-Einstein equation states that distinct diffusing populations usually arise from distinct hydrodynamic radii of the molecule or distinct local viscosity. Is Dendra2 strictly monomeric in cells? If so, then the distinct local viscosity in the nucleus seems to be the cause of distinct diffusing populations.

3) Could the DNA-binding or protein-binding properties of c-myc contribute to their distinct diffusing populations (or the relative proportions of each population)? Can a c-myc mutant defective in DNA binding be examined to test these possibilities? C-myc forms heterodimer with Max through its leucine zipper domain. Can a c-myc mutant defective in Max binding be examined?

4) The authors observed that P-TEFb, but not c-myc or free Dendra2 has anomalous diffusive behavior. FRAP or FCS can rarely differentiate multiple diffusing populations versus anomalous subdiffusion. This is one of very few reports that successfully identified anomalous diffusive behavior using single molecule imaging. How is the anomalous diffusion related to other kinetic properties observed for P-TEFb, such as abrupt transitions and biased angular distribution between consecutive steps?

5) “the time needed to find a 10nm target at a distance of 250nm is 68 times longer for c-Myc compared to P-TEFb”. Does the search time also depend on the nuclear concentration of transcription factors?

Reviewer #3:

The manuscript by Izeddin, Recamier, et al. presents a thorough study of intra-nuclear protein diffusion and concludes that two studied proteins, c-Myc and P-TEFb, exhibit different types of diffusion with c-Myc showing normal and P-TEFb anomalous diffusion. The study is timely, important to the biophysics community and beyond, and presents exciting new data that are carefully analyzed. To my knowledge, this is one the first, if not the first, study to track intra-nuclear motion of proteins at such high space and time resolution.

However, I contest author's interpretation of their data and mostly their conclusion that P-TEFb exhibits an anomalous diffusion. Below I suggest several approaches how this mechanism can be tested by simulations and data analysis.

My specific points are as follows:

Let me start by proposing an alternative model to explain P-TEFb data: labeled Cyclin-T (in complex with CDK or without) is diffusing freely while intermittently binding its partners/targets and other nuclear structures such as Pol II clusters and nuclear speckles. While bound to the targets, P-TEFb continues diffusing either with them, but much slower due to their size, or on the surface/volume of immobile targets (e.g. hopping between CTD domains of Pol II). Thus P-TEFb alternates between rounds of fast free (normal) diffusion and periods of slow and possibly confined diffusion. My hypothesis is that such motion can leads to MSD vs t and angular distributions of steps that appearas anomalous diffusion, even when the dwell time is exponentially distributed (not power-law-CTRW). Moreover, heavy-tailed dwell times can lead to real subdiffusion (CTRW type) with inhomogeneous angular distributions due to trapping.

Distinction between intermittent binding and anomalous diffusion is more than semantic. Intermittent binding can slow-down diffusion, lead to confined motion of the bound protein, but does not lead to phenomena specific to anomalous diffusion such as local exploration and distance-dependent search time. Since some conclusion of the paper rely on the statement of anomalous diffusion, it should be critically evaluated against seemingly more plausible intermittent binding.

Intermittent binding, indeed, requires fewer assumptions that anomalous diffusion. Most of homogeneous or inhomogeneous distributions of immobile or slowly moving partners/targets can lead to intermittent binding. Anomalous diffusion, in contrast, requires presence of some self-similar fractal structure present on allsales, an assumption that, in my opinion, is hard to justify.

It is likely that c-Myc and P-TEFb show different characteristics of diffusion due to different size, spatial distribution and dwell time on their targets/traps. Below I argue that presented data on P-TEFb may very well agree with the intermittent binding and not with anomalous diffusion.

1) Authors note that “individual trajectories of P-TEFb molecules often showed abrupt transitions from slow to fast displacement modes within the same trajectory”, which is consistent with the intermittent binding mechanism. Moreover, they note that “P-TEFb, the typical translocation length and the translocation histograms were comparable to those obtained for c-Myc” again consistent with intermittent binding. My guess is that subdiffusion on a fractal (i.e in the presence of a fractal-distributed traps) leads to power-law distributed displacements. Authors can test this for their simulations of anomalous diffusion on a percolation cluster.

My suggestion is to develop simulations where a diffusing molecule moves freely and gets trapped into finite size traps (containers, e.g. speckles or Pol II clusters) inside which a molecule can also move and then escape after some dwell time.

2) Fit of t^a of the MSD vs t is not very convincing. (a) MSD/t curves for c-Myc or Dentra are not flat either. For c-Myc and t>0.03s MSD/t vs t points easily fall onto a straight line. This reflecta either some real biophysical effect that affect both c-Myc and P-TEFb or some issues with longer trajectories and/or trajectory selection biases. Either way, the only difference between c-Myc and P-TEFb curves are in the first three points. (b) Most importantly, intermittent binding may very well create such “anomalous-looking” MSD/t vs t plots. To test this, authors can use simulations of the intermittent binding I suggested above, simulate the same length and number of trajectories as in the experiment and test whether they indeed can produce such results. Sweeping parameters of the intermittent simulations to fit the data may be necessary. Such parameters include the mean size of a trap, the number of traps (assuming a homogeneous distribution), and the mean dwell time (assuming exp distribution).

Another way to test for anomalous diffusion vs intermittent binding would be to segment trajectory into fast and slow parts and analyze them separately, perhaps by collapsing slow parts into points. Some steps toward this have been done by removing immobile steps (Figure 4—figure supplement 1), but a more systematic segmentation can be done (e.g. by applying HHM to the time series of step sizes). My guess is that true anomalous diffusion should manifest itself in power-law distributed step sizes and the same MSD∼t scaling for all time scales. Intermittent binding, on the contrary is expected to show normal diffusion for fast phases and confined diffusion (MSD going into a plateau for larger t) for slow phases.

3) The angular distribution of consecutive steps observed for P-TEFb is not a very strong argument in support of anomalous diffusion. In fact, enrichment of trajectory reversals (90-180deg) can be observed for trapped particles. This is evident in the angle distribution of H2B, which is a mixture of trapped and freely moving proteins. As such, observed angular distribution for P-TEFb may very well reflect its trapping/confinement during which the protein either fluctuates at one place or moves within a small volume, thus making sharp reversals. Simulations and analysis of trajectories that I suggested above can help to answer test this possibility.

Angular distributions for steps separated by delta_t don't seem to support anomalous diffusion of P-TEFb either. Comparison on these distributions for experiments (Figure 3) and simulated anomalous diffusion (Figure 4) shows that H2B is in best agreement with simulated anomalous diffusion. This argument only reinforces my concern that these plots cannot distinguish anomalous diffusion and a mixture of immobilized and freely-diffusing trajectories.

Moreover, when immobile steps are removed (Figure 4—figure supplement 1), angular distributions for P-TEFb and c-Myc looks very much alike, with both proteins showing enrichment of reversals for delta_t > 40ms. Speaking of c-Myc, authors rightfully note that this may reflect “confinement to domains significantly smaller than the nucleus”. The same argument can be equally applied to P-TEFb. These distributions for c-Myc and P-TEFb differ for ∼10-20ms range, possibly reflecting differences in sizes of traps and dwell times. By sweeping parameters for simulations that I proposed above one can find size/dwell times consistent with the data for each protein.

4) In Discussion, authors mention some important experimental results that they plan to publish elsewhere. They mention that impediment of interactions between P-TEFb and Pol II leads to a change in P-TEFb diffusion from anomalous to normal. In my opinion, this is very important result and the paper would be much stronger if it were presented here. Authors further suggest that a matrix of Pol II-CTD repeats can lead to anomalous diffusion. This is a conceptually important point: a mesh of traps can lead to slow diffusion, diffusion with intermittent binding, but anomalous diffusion would further require such mesh of Pol II-CTDs to form a perfect fractal. Note that anomalous diffusion can be observed on the percolation cluster only right at the percolation threshold. Near-fractal clusters below or above the percolation point do not lead to anomalous diffusion. It is hard to imagine Pol II forming such perfect structures. Excellent recent data on Pol II localization (from the same group) would hardly support this notion.

5) As far as simulations are concerned, simulations used to test possible modes of diffusion are important and insightful. I wasn't however that much impressed by simulations of the search process by normal and anomalous diffusion (Figure 5). Very similar results for search by local vs non-local explorers can be found in other papers. I also found surprising the set-up of the simulations: one molecule looking for a single target in the nucleus. Given the number of molecules per nucleus the search can almost instantaneous.

Here is my argument. The number of molecules of c-Myc per cell is ∼10^5 (bionumbers.org), which exceeds ∼10^4 c-Myc targets. The number of active Pol II, i.e. those that have P-TEFb bound, can also be estimated as ∼10^5-10^6 per cell. Thus in 500um^3 of the nuclear volume the spacing between c-Myc molecules and the spacing between P-TEFb is of the order of ∼100nm, i.e. any target has a protein within 100nm. As evident from MSD data, the area of (100nm)^2 is swept by either protein in less than 10ms, suggesting that the search time should be of the order of ∼10ms, irrespective of the mode of diffusion.

To summarize, my recommendation to the authors is to revaluate their conclusions concerning anomalous diffusion and, at least, to consider intermittent binding as an alternative mechanism. Suggested analysis and simulation may allow to estimate sizes of targets and dwell times for c-Myc and P-TEFb. I believe this analysis can only strengthen this solid and important paper.

---

## [Author Response]

We are pleased to resubmit a revised version of our work “Distinct target search modes for c-Myc and P-TEFb revealed by single molecule tracking in live cells”. We have taken into consideration the comments from the reviewers and we think that we have considerably improved our manuscript.

There are three major improvements in this revised paper:

1) We have reanalyzed our data, in particular the antipersistent characteristics of the molecules traces. We have defined an “asymmetry coefficient” (AC) that allowed us to study the features of the angular distribution at several temporal and length scales.

2) We have tested the alternative models proposed by Reviewer 3. While the simplest intermittent models were insufficient to fully reproduce the characteristics of our data, a slightly more sophisticated model involving intermittent diffusion and trapping replicated the behavior of c-Myc. Also, a model of intermittent trapping invoking a hierarchy of trap sizes distribution naturally leads us to considering the fractal organization of the nuclear space. We feel that these scenarios have significantly enriched our paper without substantial changes of our initial conclusions.

3) We have improved the narrative flow of our story, as suggested by Reviewer 1. We have rewritten the Introduction and include considerations about the influence of crowding on molecular mobility and reaction rates. We have also introduced earlier the concepts of compact and *non-compact* exploration. Finally, we have significantly reduced the details about the experimental set-up and the initial analysis of Dendra2 and H2B.

Please, find below the detailed answer to the complete reviews.

Reviewer #1:

*[…] Specifically*:

*1) I wish the key concept, of compact versus non-compact exploration of space, was introduced earlier and used to actively guide the narrative. As the text is currently written, the reader is exposed to a lot of data that is pretty standard – SPT trajectories and their analysis as diffusive motion – before he/she encounters the novel aspects of the work. This burying of the lead is against the authors' own interest, I think*.

We have rewritten the Introduction in order to include the concept of compact and *non-compact* exploration earlier in the manuscript. We have also included considerations about the consequences of sampling and different modes of exploration on biochemical reaction rates. We have greatly reduced the details about the initial conventional analysis of the SPT. However, we think this previous step is necessary in order to convey the necessity of novel approaches of data-analysis other than simple MSD, which is now a day the overwhelming standard in SPT experiments.

*2) While an overabundance of optics and imaging details are provided in the main text, essential biological details are missing. For example, it is not satisfactorily described how the fusion TFs were expressed, and what evidence there is that the fusions behave like the unfused wild type. The authors “tested the functionality of c-Myc-Dendra2 by performing RT-qPCR on a set of active genes in our U2OS cell line”. Was the fusion expressed in the null background? From the endogenous promoter? Does the fusion protein rescue the null? And as for the other fusion, CycT1-Dendra2, we are pointed to a work “in preparation” and are thus unable to judge for ourselves the evidence for its functionality*.

We have rewritten the details about the c-Myc-Dendra2 fusion in the Materials and Methods section. For the c-Myc experiments, cells were transfected with c-Myc-Dendra2 and cells with a very low over-expression of c-Myc-Dendra2 were used for the experiments on the microscope. This was not a stable cell line but a transitory expression of c-Myc-Dendra2. Our RT-qPCR assay showed an increase activation of genes when compared to wild type, untransfected cells. This is in line with previous works on c-Myc (57; 42), notably Nie and coworkers using also the C terminal of c-Myc fused to Dendra2.

We have removed all references to our other work in preparation. However, in this work we have measured the functionality of P-TEFb, its interactions with the CTD of RNA Pol II and its regulation by a set of techniques (FRAP, FLIP, SPT...), which would be difficult to summarize in one figure.

*3) The presentation of experimental data is also less than optimal. Examples: (a) Sample sizes (# trajectories, # cells, etc.) need to be stated for each plot, but they are not. (b) The evidence for sub-diffusion (*Figure 3*) is not convincing. The anomalous behavior of MSD should be demonstrated for individual trajectories, not for an ensemble average. (c) The evidence for time-evolution of angle preference (*Figure 4*) is also unconvincing. A single parameter should be extracted from the plot at each time point, and the change of this parameter value over time should be examined to reveal whether the angle preference changes over time or not. This would allow direct comparison to theory (*Figure 5*)*.

(a) We have included details about the number of cells and trajectories in the figures legends.

(b) Figure 3 illustrates the problematic of considering the MSD as the sole indicator of anomalous diffusion. In particular with a 3D movement of a mixed diffusive population of molecules recorded with a limited depth of focus (population exclusion effect). However, the good fit to a power law of the MSD curve of P-TEFb, in combination with the remarkably good collapse of the cumulative histograms (Figure 3—figure supplement 3) is the best indication for anomalous diffusion that one can obtain with this classical analysis given the experimental limitations. The “indication” of anomalous diffusion for P-TEFb is later confirmed with the new analysis of the angular distribution and its temporal evolution. Note that our traces are extremely short due to the fast mobility of the molecules and a limited (∼1um) depth of focus. When molecules exit the depth of focus the trace is stopped, in order to avoid reconnection errors. That is the reason why we can only plot averaged MSD and not MSD of the individual trajectories for most of our data.

(c) We have now reanalyzed the antipersistence of our traces reflected in the angular histograms. We have defined an asymmetry coefficient (AC) as the logarithm to the base 2 of the ratio between the frequencies of forward angles (between 0° and 30°) and the backward angles (150° - 180°) (new Figure 4). The AC is thus negative for angular distributions with a dominant number of backward angles and it measures the deviation from a homogenous distribution. We have use such an AC to study the temporal and spatial dependence of the angular distribution (new Figure 4 and Figure 4—figure supplement 1) and used it to test the different proposed models (new Figure 5 and Figure 5—figure supplement 2).

Reviewer #2:

*[…] Specific points*:

*1) The actual data presenting the described results (diffusion coefficients, etc) cannot be found*.

The diffusion coefficients and parameters are obtained from the 3-population fit of the translocation histograms (new Figure 2—figure supplement 1) following the procedure described in Analytical Methods III.

*2) It is interesting that free Dendra2 and c-myc have several populations of very distinct diffusion coefficients. Stokes-Einstein equation states that distinct diffusing populations usually arise from distinct hydrodynamic radii of the molecule or distinct local viscosity. Is Dendra2 strictly monomeric in cells? If so, then the distinct local viscosity in the nucleus seems to be the cause of distinct diffusing populations*.

Dendra2 has been shown to be monomeric in HeLa cells (31). We agree with this interpretation, the different viscosity/crowding of different nuclear regions is most likely the cause of the different diffusing populations. Similar conclusions have been withdrawn with inert probes like streptavidin (30) and quantum dots (3). We have included a statement in the Conclusion. It is also not possible to exclude that Dendra2 interacts with a weak affinity to nuclear components.

*3) Could the DNA-binding or protein-binding properties of c-myc contribute to their distinct diffusing populations (or the relative proportions of each population)? Can a c-myc mutant defective in DNA binding be examined to test these possibilities? C-myc forms heterodimer with Max through its leucine zipper domain*. *Can a c-myc mutant defective in Max binding be examined?*

We agree with the interpretation of Reviewer 2. We believe that the diffusive behavior of c-Myc is a reflection of its interaction with a multiplicity of partners including Max. Dynamics of c-Myc and Max have been investigated by Fluorescent Recovery After Photo-bleaching (FRAP) (Phair et al, Methods Enzymol, 2004), which provides reliable estimates for different diffusive population. With our SPT approach, we have however focused on the geometry of exploration that has led us to the observation of both, compact and *non-compact* explorations. Since c-Myc is a *non-compact* explorer, it is difficult to imagine that a mutant defective in DNA binding would switch to a compact mode of exploration.

*4) The authors observed that P-TEFb, but not c-myc or free Dendra2 has anomalous diffusive behavior. FRAP or FCS can rarely differentiate multiple diffusing populations versus anomalous subdiffusion. This is one of very few reports that successfully identified anomalous diffusive behavior using single molecule imaging*. *How is the anomalous diffusion related to other kinetic properties observed for P-TEFb, such as abrupt transitions and biased angular distribution between consecutive steps?*

The “abrupt transitions” mentioned in the previous version of the manuscript were difficult to quantify and hence somewhat misleading. We have therefore removed this description from the resubmitted version of the paper.

The asymmetric angular distribution, on the other hand, is a signature of the complex dynamics of P-TEFb and a valid parameter to evaluate the origin of its diffusion mode, giving rise to anomalous diffusion. We have extended our analysis of this angular distribution by defining an asymmetry coefficient (AC) (Figure 4).

*5) “the time needed to find a 10nm target at a distance of 250nm is 68 times longer for c-Myc compared to P-TEFb”. Does the search time also depend on the nuclear concentration of transcription factors*?

This is the time for a single TF, with the parameters that we measured. One needs to take into consideration the number of molecules in the nucleus (in the order of 10^5) and hence the target search is a “parallel” search. The times provided in Figure 6 are thus to be taken as relative rather than absolute. The importance of this model, rather than the target search time, is the increased probability of redundant interactions with the same sites (or cluster of sites) in the compact mode, as opposed to the *non-compact* exploration.

Reviewer #3:

*[…] My specific points are as follows*:

*Let me start by proposing an alternative model to explain P-TEFb data: labeled Cyclin-T (in complex with CDK or without) is diffusing freely while intermittently binding its partners/targets and other nuclear structures such as Pol II clusters and nuclear speckles. While bound to the targets, P-TEFb continues diffusing either with them, but much slower due to their size, or on the surface/volume of immobile targets (e.g. hopping between CTD domains of Pol II). Thus P-TEFb alternates between rounds of fast free (normal) diffusion and periods of slow and possibly confined diffusion. My hypothesis is that such motion can leads to MSD vs t and angular distributions of steps that appearas anomalous diffusion, even when the dwell time is exponentially distributed (not power-law-CTRW). Moreover, heavy-tailed dwell times can lead to real subdiffusion (CTRW type) with inhomogeneous angular distributions due to trapping*.

*Distinction between intermittent binding and anomalous diffusion is more than semantic. Intermittent binding can slow-down diffusion, lead to confined motion of the bound protein, but does not lead to phenomena specific to anomalous diffusion such as local exploration and distance-dependent search time. Since some conclusion of the paper rely on the statement of anomalous diffusion, it should be critically evaluated against seemingly more plausible intermittent binding*.

*Intermittent binding, indeed, requires fewer assumptions that anomalous diffusion. Most of homogeneous or inhomogeneous distributions of immobile or slowly moving partners/targets can lead to intermittent binding. Anomalous diffusion, in contrast, requires presence of some self-similar fractal structure present on allsales, an assumption that, in my opinion, is hard to justify*.

*It is likely that c-Myc and P-TEFb show different characteristics of diffusion due to different size, spatial distribution and dwell time on their targets/traps. Below I argue that presented data on P-TEFb may very well agree with the intermittent binding and not with anomalous diffusion*.

We are thankful to Reviewer 3 for his detailed review of our manuscript and subsequent remarks. As Reviewer 3 has pointed out, the distinction between normal diffusion and subdiffusion is essential for the interpretation of our data and it has previously been warned that single particle experiments can be misinterpreted in this respect (Martin et al., 2002). Moreover, Reviewer 3 has suggested alternative simpler models that can be tested against our experimental data.

In order to fully investigate these suggestions, we have reanalyzed our data and extracted an “asymmetry coefficient” that parameterizes the deviation of homogeneity of the angular distribution. The temporal evolution of this asymmetry coefficient and its dependence with the average length between consecutive translocations provides information about the origin of the asymmetry (Figure 4 and Figure 4—figure supplement 1 of the resubmitted manuscript). Using these parameters, we have been able to test the validity of the different models proposed by Reviewer 3, i.e. intermittent fast/slow diffusion, and an intermittent trapping of the molecule in traps of a fixed size or a distribution of sizes.

Our conclusion is that the simplest intermittent models were insufficient to fully reproduce the characteristics of our data. However, a model involving intermittent diffusion and trapping replicated the behavior of c-Myc. Also, a model of intermittent trapping with a power law distribution of trap sizes is able to partly reproduce the results obtained for P-TEFb. Nonetheless, we argue that invoking a hierarchy of size distribution corresponds to our fractal hypothesis.

Below there is a detailed description of the simulations suggested by Reviewer 3.

We have parameterized the asymmetry found in the angular distribution of the data. We call this the “asymmetry coefficient” (AC), which is the logarithm to the base 2 of the ratio between the frequencies of forward angles (between 0° and 30°) and the backward angles (150° - 180°) (Figure 4). The AC is thus negative for angular distributions with a dominant number of backward angles and it measures the deviation from a homogenous distribution.

We have explored the dependence of the AC with time and space. We calculated the AC for the angles formed at increasing lag times as well as a function of the average length of the consecutive steps forming the angle θ (Figure 4). Please, not that the first data point of the spatial dependence of the AC is always reflecting the experimental localization accuracy.

We observed that for c-Myc (and also for Dendra2, as shown in Figure 4—figure supplement 1) the AC decreases with time. We interpret this as a confined diffusion in which the borders of the trapping volume are seen only at larger times (the volume is big compared to the translocation step). The spatial dependence of the AC for c-Myc and Dendra2 evolves from a negative value (due to the localization precision) to a homogenous distribution (AC ≈ 0). On the other hand (with the exception of the first spatial bin, biased by the experimental localization accuracy) P-TEFb shows a constant AC in both time and space. (All the angular histograms and the data for the other proteins are shown in Figure 4—figure supplement 1.)

With these parameters in mind, we have tested a set of simulations. First, an intermittent fast / slow diffusion model. Then, intermittent binding to a trap of a given radius. Finally, intermittent binding to traps with a distribution of sizes.

The parameters for the fast/slow model were the following: “fast” diffusion coefficient D1, “slow” diffusion coefficient D2, probability Kon of transition from D1 to D2, and probability Koff of transition from D2 to D1. In order to obtain an outcome that resembles our experimental data, we determined D1 and D2 from a two-exponential fit of the cumulative translocation histograms. The ratio between the two populations also determined the ratio Kon/Koff, which we explored in order to obtain a translocation histogram as close as possible to that obtained in the experiments.

Similarly, we determined the best parameters for the intermittent trap model with a fixed trap radius, whose parameters are the diffusion coefficient D, the probability Kon to enter a trap, the probability Koff to escape the trap, and the radius of the trap Rtrap.

With these constrains, we obtained a relatively good likeness between the translocation histograms of c-Myc and P-TEFb, and the outcome of the simulated traces (Figure 5—figure supplement 1).

In Figure 5—figure supplement 2, we have plotted the results for the parameters that best fitted c-Myc and P-TEFb. With the fast/slow diffusion model, we retrieved mild negative values of the AC. These were prominently at small lag times and small translocation steps. The tendency for larger times and steps was always toward AC = 0, that is a symmetric angular distribution.

We obtained stronger asymmetries with the intermittent trap model. These curves also evolve with time and space, and are strongly dependent of the size of the trap. The simulated particle undergoes confined diffusion naturally only when it is inside the trap.

Although we reproduced certain characteristics of the angular distribution of our data, these simple simulations failed to reproduce the temporal and spatial scaleless behavior of P-TEFb.

We then considered a model that results from a combination of intermittent diffusion and intermittent trap. We performed simulations of fast diffusing particles with a probability to engage into a slower diffusion confined in a trap (Figure 5). Here, AC decreased with increasing lag times (Figure 5), reproducing the trend observed in c-Myc. Likewise, AC displayed the same behavior as c-Myc, tending to zero for larger values of the translocation steps (Figure 5). Following this model, c-Myc performs thus a free exploration of the nuclear space, combined with slower yet still normal diffusion of confined domains, reflecting its interactions with a multiplicity of partners.

We then moved on to a model of intermittent trapping with a distribution of trap sizes. We tested an exponential (not shown) as well as a Pareto (power law) distribution. The extra parameters were the mean radius of the trap (for the exponential distribution) and a minimum trap size and exponent for the power law distribution.

This model reproduced better the temporal dependence of the AC of P-TEFb. Although the AC of the simulations was not strictly constant with time it does not show a tendency towards zero (Figure 5). Also, the spatial dependence plateaus after a given translocation step (Figure 5). (The first point in the graph of is to be ignored, as it is a manifestation of the simulated localization accuracy.)

These set of simulations show that indeed, the anticorrelations that we observed with P-TEFb could indeed be due to confinement, but that is necessary to invoke a certain degree of hierarchical organization of confining domains. This is precisely what leads us to hypothesize a fractal organization of the available space. Our data show that for PTEFb successive steps are anti-correlated at all observed time scales (Figure 4), but also at all observed length scales (Figure 4). This gives clear evidence that correlations exist over the whole observed range, which is compatible with a fractal like organization of the accessible space. We agree that confinement also leads to anti correlations: these however exist only at the scale of the confining domain. To explain our observations, one should therefore invoke a hierarchy of sizes of confining domain, which corresponds to our fractal hypothesis.

We thus explored different fractl networks: the 3D percolation cluster already simulated in the previous version of the manuscript, and as a 2D Sierpinski carpet. For both fractal structures the angular anisotropy was constant with time (Figure 5), illustrating the scale-invariant features of fractal structures, as observed in the experimental data of P-TEFb. Surprisingly, the AC decreased for larger translocations in the case of the percolation cluster, while the Sierpinski carpet yielded an invariant asymmetry in space. This was interesting because it indicates that the underlying network needs to reproduce a certain degree of geometrical self-similarity, as it is the case of the Sierpinski carpet. The percolation cluster, on the other hand, does not conserve its geometry at different scales but rather other features like the local density obeys a power law over several length scales.

*1) Authors note that “individual trajectories of P-TEFb molecules often showed abrupt transitions from slow to fast displacement modes within the same trajectory”, which is consistent with the intermittent binding mechanism. Moreover, they note that “P-TEFb, the typical translocation length and the translocation histograms were comparable to those obtained for c-Myc” again consistent with intermittent binding. My guess is that subdiffusion on a fractal (i.e in the presence of a fractal-distributed traps) leads to power-law distributed displacements. Authors can test this for their simulations of anomalous diffusion on a percolation cluster*.

*My suggestion is to develop simulations where a diffusing molecule moves freely and gets trapped into finite size traps (containers, e.g. speckles or Pol II clusters) inside which a molecule can also move and then escape after some dwell time*.

*2) Fit of t^a of the MSD vs t is not very convincing. (a) MSD/t curves for c-Myc or Dentra are not flat either. For c-Myc and t>0.03s MSD/t vs t points easily fall onto a straight line. This reflecta either some real biophysical effect that affect both c-Myc and P-TEFb or some issues with longer trajectories and/or trajectory selection biases. Either way, the only difference between c-Myc and P-TEFb curves are in the first three points. (b) Most importantly, intermittent binding may very well create such “anomalous-looking” MSD/t vs t plots. To test this, authors can use simulations of the intermittent binding I suggested above, simulate the same length and number of trajectories as in the experiment and test whether they indeed can produce such results. Sweeping parameters of the intermittent simulations to fit the data may be necessary. Such parameters include the mean size of a trap, the number of traps (assuming a homogeneous distribution), and the mean dwell time (assuming exp distribution)*.

Under our experimental conditions, the MSD curves of a mixed population of diffusive molecules are affected by a “population exclusion” effect, due to different defocusing rates (Analytical Methods III, and Figure 2—figure supplement 2). We indeed found this effect in the intermittent diffusion simulations. The MSD as a unique tool to discern normal versus anomalous diffusion is thus not a good indicator in these experiments. What lead us to the anomalous hypothesis for P-TEFb was 1) The good fit of the MSD with an exponential law (Figure 3), 2) The good collapse of the cumulative histogram of translocations (Figure 3—figure supplement 3), and 3) the further confirmation of our hypothesis with the analysis of the asymmetry coefficient, validated with simulations.

*Another way to test for anomalous diffusion vs intermittent binding would be to segment trajectory into fast and slow parts and analyze them separately, perhaps by collapsing slow parts into points. Some steps toward this have been done by removing immobile steps (*Figure 4—figure supplement 1*), but a more systematic segmentation can be done (e.g. by applying HHM to the time series of step sizes). My guess is that true anomalous diffusion should manifest itself in power-law distributed step sizes and the same MSD∼t scaling for all time scales. Intermittent binding, on the contrary is expected to show normal diffusion for fast phases and confined diffusion (MSD going into a plateau for larger t) for slow phases*.

With the short traces obtained in our experiments, we are not in favor of this methodology because it involves some sort of thresholding, possibly leading to an artificial separation of populations.

3) The angular distribution of consecutive steps observed for P-TEFb is not a very strong argument in support of anomalous diffusion […]

See above.

Angular distributions for steps separated by delta_t don't seem to support anomalous diffusion of P-TEFb either […]

See above.

*Moreover, when immobile steps are removed (*Figure 4—figure supplement 1*), angular distributions for P-TEFb and c-Myc looks very much alike […]*

See above.

*4) In Discussion, authors mention some important experimental results that they plan to publish elsewhere. They mention that impediment of interactions between P-TEFb and Pol II leads to a change in P-TEFb diffusion from anomalous to normal. In my opinion, this is very important result and the paper would be much stronger if it were presented here. Authors further suggest that a matrix of Pol II-CTD repeats can lead to anomalous diffusion. This is a conceptually important point: a mesh of traps can lead to slow diffusion, diffusion with intermittent binding, but anomalous diffusion would further require such mesh of Pol II-CTDs to form a perfect fractal. Note that anomalous diffusion can be observed on the percolation cluster only right at the percolation threshold. Near-fractal clusters below or above the percolation point do not lead to anomalous diffusion. It is hard to imagine Pol II forming such perfect structures. Excellent recent data on Pol II localization (from the same group) would hardly support this notion*.

We have removed the references to unpublished work. Although we agree that this supports our conclusions, the focus of this work is to elucidate the coexistence of different modes of nuclear diffusion in the nucleus, and the consequences for biochemical rates. New data about the interaction between P-TEFb and the CTD of RNA Pol II, as well as its regulation by the 7SK snRNP comprises, SPT, FRAP, and FLIP experiments, and it would be impossible to summarize in one figure.

*5) As far as simulations are concerned, simulations used to test possible modes of diffusion are important and insightful. I wasn't however that much impressed by simulations of the search process by normal and anomalous diffusion (*Figure 5*). Very similar results for search by local vs non-local explorers can be found in other papers. I also found surprising the set-up of the simulations: one molecule looking for a single target in the nucleus. Given the number of molecules per nucleus the search can almost instantaneous*.

*Here is my argument. The number of molecules of c-Myc per cell is ∼10^5 (**bionumbers.org**), which exceeds ∼10^4 c-Myc targets. The number of active Pol II, i.e. those that have P-TEFb bound, can also be estimated as ∼10^5-10^6 per cell. Thus in 500um^3 of the nuclear volume the spacing between c-Myc molecules and the spacing between P-TEFb is of the order of ∼100nm, i.e. any target has a protein within 100nm. As evident from MSD data, the area of (100nm)^2 is swept by either protein in less than 10ms, suggesting that the search time should be of the order of ∼10ms, irrespective of the mode of diffusion*.

The simulations shown in Figure 6 show indeed the mean first passage time for one single molecule under consideration. The total number of c-Myc and P-TEFb molecules in the nucleus is indeed in the order of 10^5. The search process is therefore a “parallel” one and the times obtained from our simulations have to be taken as relative and not absolute values.

We think that the strength of Figure 6 lies on the illustration of how interaction with one particular target can be favored in respect to another (in the case of compact exploration). Also, how in a compact mode of exploration the probability of revisiting a target is larger than visiting a new one. These considerations are true irrespectively of how fast the target-search is, and are important for the understanding of the nuclear architecture and its influence on regulation of gene transcription.

*To summarize, my recommendation to the authors is to revaluate their conclusions concerning anomalous diffusion and, at least, to consider intermittent binding as an alternative mechanism. Suggested analysis and simulation may allow to estimate sizes of targets and dwell times for c-Myc and P-TEFb. I believe this analysis can only strengthen this solid and important paper*.

We acknowledge to Reviewer 3 the pertinence and importance of his criticisms and suggestions. We believe that the study of the alternative models proposed by Reviewer 3 have greatly strengthen our work, in particular to reproduce the diffusive behavior of c-Myc as well as to a better argumentation towards our fractal model for P-TEFb.